# Methods for Estimating Avoidable Costs of Excessive Alcohol Consumption

**DOI:** 10.3390/ijerph18094964

**Published:** 2021-05-07

**Authors:** Beata Gavurova, Miriama Tarhanicova

**Affiliations:** 1Faculty of Management and Economics, Tomas Bata University in Zlín, 760 00 Zlín, Czech Republic; 2Faculty of Mining, Ecology, Process Control and Geotechnologies, Technical University of Košice, 042 00 Košice, Slovakia; miriama.tarhanicova@tuke.sk

**Keywords:** cost of illness, alcohol addiction, lost productivity, Czech Republic

## Abstract

*Background*: Alcohol is a risk factor with serious consequences for society and individuals. This study aims to present methods and approaches that might be used to estimate the costs related to excessive alcohol consumption. It emphasizes the need for general methods and approaches that are easily applicable, because the level of digitalization and data availability vary across regions. The lack of data makes many methods inapplicable and useless. The ease of applicability will help to make cost-of-illness studies and their results comparable globally. *Methods*: This study is based on data from the Czech Republic in 2017. Drinking alcohol results in costs of healthcare, social care, law enforcement, and administrative costs of public authorities. To quantify the cost of drinking in the Czech Republic, the top-down approach, bottom-up approach, human capital approach and attributable fractions were used. *Results*: In 2017, the cost related to alcohol was estimated at 0.66% of the national GDP. Lost productivity represented 54.45% of total cost related to alcohol. All cost related to alcohol is considered to be avoidable. *Conclusions*: The methods and approaches applied to estimate the cost of disease or any other health issue should be generalized regarding the availability of data and specifics of provided services to people who are addicted or have any kind of disability.

## 1. Introduction

Alcohol is a risk factor, and its consumption may result in several diagnoses, economic burden, lost productivity, and avoidable deaths. It causes harms not only to its users, but also indirectly to its non-users because they might be in social networks with the addicted. There are people who live with addicted persons and might be impacted by their addiction financially, psychically, and physically, because drunk people endanger others in many ways (conflicts, sexual abuse, criminal activity, debts, etc.). Moreover, people under the influence of alcohol might cause property damage. One dangerous phenomenon is driving under the influence of alcohol when there are many innocent people around.

The consequences of excessive alcohol drinking bring many requirements to the treatment programs for addicts, the fights against its negative effects, and future prevention programs as well. The epidemiology of alcohol and its consequences differs across the world because of different standards of living, heterogenous economic levels, social inequalities, and health issues.

Many authors have aimed to evaluate the economic consequences of drinking alcohol in the so-called cost-of-illness (alcohol) studies. Those types of studies differ because of their heterogenous structures, types of cost included, data used, or the examined level of alcohol consumption. According to the extensive study of Onukwugha et al. [1], who reviewed 993 cost-of-illness studies, authors usually evaluate the cost related to many types of diseases such as infection and skin diseases, cancer, mental disorders, vision and hearing damage, cardiovascular diseases, respiratory diseases, disorders of digestive, circulatory and immune system, external injuries, etc. In the study of Lozano et al. [2], the authors were fully focused on estimations of the cost of infection and endocrinal diseases. Cost related to mental disorders was estimated by Angelis et al. [3]. The authors Mangen et al. [4] and Gabbe et al. [5] were focused on the country’s expenditure related to disorders of circulatory and digestive system. Jo discussed the cost of illness as a consequence of drug consumption [6].

Epidemiology of alcohol drinking becomes important because alcohol considerably raises the economic burden. As stated by Jarl et al. [7], total cost of alcohol is avoidable and can be eliminated by appropriate legislation changes and restrictions. 

Many authors, such as Currie et al. [8] or Farnham [9], take into consideration only cost related to healthcare, while other types of cost, such lost productivity or the cost of law enforcement are not included. Anders et al. [10] and Single et al. [11] distinguished direct, indirect, and intangible costs within each cost category. Currie et al. [8] confirmed that direct costs represent all direct payments related to illness or addiction, while indirect costs are those that reduce available resources. Indirect costs thus represent the costs associated with lost productivity related to morbidity (lost productivity due to sick leave) and mortality (lost productivity due to premature death as a result of addiction or any kind of disease). Farnham [9] pointed out that the cost categories in cost-of-illness studies differ significantly.

According to Jo [6], researchers should consider direct healthcare costs (institutional care, outpatient, home care, pharmacy service, ancillary services, diagnostic tests), direct non-medical costs (social service costs, program efficiency evaluations, technology-related overheads, variable costs, legal costs, travel costs, childcare), and indirect costs (lost productivity, deterioration, lost leisure time, time spent to visit members in medical facility, or psychological counselling centers, etc.). In the study of Tajima-Pozo et al. [12], the authors partly agreed with Jo [6], and divided social costs into the following groups: direct cost (concerning private physiotherapy, traditional alternative medicine, Chinese medicine, hospitalization, laboratory tests, diagnostics, support staff, prescription drugs), indirect cost (lost productivity) and intangible cost. Thavorncharoensap et al. [13] divided cost-of-illness into direct costs, (including healthcare costs, research and prevention costs, criminal and litigation costs, property damage costs or losses, administrative costs, costs of social assistance or social work, costs of procuring alcoholic beverages), other costs, and indirect costs (costs of premature mortality, costs of reduced productivity, which include the cost of lost productivity due to sick leave, and presenteeism costs of imprisonment, costs of job loss or early retirement, costs related to criminal activity, etc.). In their cost-of-illness study, Anders et al. [10] considered only material costs (loss of production capacity, calculated based on human capital approach) and intangible costs such as psychological consequences borne by victims of drug abuse, etc.

There are many methods used to estimate the cost of illness. According to French et al. [14], averting behavior methods might be used to estimate the cost because they represent an extension of more complex economic models that describe and predict household production as well. The aim is to determine whether individuals have averted behavior in order to reduce the risk of recurrence of the disease. These methods support the estimation of economic benefits of public policies, especially those that reduce morbidity or mortality. The maximization of utility is based on observable behavior. As stated by Um et al. [15], the averting behavior methods include the concept of utility, expressing a certain degree of satisfaction, and pleasure.

Another method used to estimate cost of health issues is the consumer market method. As stated by French et al. [14], Um et al. [15], Gold et al. [16], and Ippolito et al. [17], the consumer market method was originally used to determine whether the market is able to respond to changes in security risks. It has been applied in studies that deal with various safety features such as seatbelts, smoke detectors, or information on the harmfulness of smoking. To estimate lost productivity, Rothermich et al. [18], Kishore et al. [19], Koopmanschap et al. [20], and Brouwer et al. [21] applied the friction cost method. According to their studies, the friction cost method is the most appropriate method for estimating indirect costs such as the lost productivity. The value of indirect costs is approximated by the average income of individuals in the future. Thus, this friction period represents the period during which productivity of one worker is lost due to a disease (or death) and the company is looking for the worker’s replacement. However, they also admit that the friction cost method is applied in the studies of primary data, at the level of enterprises, which examine the cost related to lost productivity of individuals. Therefore, they do not recommend the friction cost method for estimating costs in macroeconomic studies.

According to Jo [6], the human capital approach is designed to estimate the value of human capital. Human capital is perceived as a labor force that actively contributes to GDP in the economy. This value of human capital represents the present value of future income, assuming that future income will be used as a substitute for future productivity. When calculating, this approach requires calculating the present value of future income. The choice of an appropriate discount rate is very important, with a value usually between 2% and 3%. Calculation of intangible cost, as presented by Anders et al., should be based on a willingness-to-pay approach [10].

Recent research of Jarl [22] showed that the demographic approach is complementary to the human capital approach. It compares the current size and structure of the population with the size and structure of the hypothetical population in the world, where the observed phenomenon does not occur (such as drug use, diabetes, etc.). It thus determines the cost of the sacrificed opportunity. The demographic approach estimates the value of a loss over one year of life. Le et al. [23] state that the human capital approach has many forms: the income-based approach, output-based approach, cost-based approach, indicator-based, Human Development Index, and volume indices. Those forms are applied according to the available data resources.

The next method which might be used in cost estimation is the contingent valuation method. Hodgson and Meiners [24] state that the willingness-to-pay approach does not estimate the components of cost resulting from disease. It proposes that the value of health can be deducted from the amount people are willing to pay to reduce the probability of an event (death or illness). In general, it is a method for estimating the value of statistical life (VSL).

The willingness-to-pay approach (WTP) is an alternative to the previous two approaches. Instead of traditional disease valuation, WTP is an approach that is based on people’s readiness to pay to avert any kind of danger such as disease, environmental catastrophe, or the side effects of smoking or drinking alcohol. It introduces the concept of utility in the cost calculation.

According to study of Dickie [25], there are four obstacles of behavior-averting methods that complicate the measurement of cost. Those obstacles are joint production, badly measured price of averting methods, difficulty of identifying the partial effects of averting behavior and of environmental conditions, and difficulty of identifying the total effect of environmental conditions. Hodgson and Meiners [24] and Breidert et al. [26] state that it is difficult to apply WTP, and therefore it is not frequently applied in COI studies. However, French et al. state [14] that the WTP is applied mostly in the studies of averting behavior. According to Cookson [27] and Gafni [28], this approach can be applied to secondary data or data from laboratory experiments, or in the case of studying the behavior of individuals, e.g., during auctions. It is also applied in the case of primary data collection, surveys (direct and indirect).

Hodgson and Meiners [24] criticize the human capital approach for the lack of a conceptual framework of a measurement of an economic value of life. The human capital approach has been criticized by other authors, such as Jo, because it does not take into account the social status of patients and thus does not provide an attitude towards individual social groups; it is a summary for the whole population [6]. Compared to the WTP, the advantage of the human capital approach is that it does not require extensive questionnaires to determine the preferences of the population, and therefore it is not tied to hypothetical questions that might arise in questionnaires. However, Pagano et al. [29] and Pike and Grosse [30] argue that the human capital approach does not take into account absence (due to morbidity or mortality) which may be filled by another employee, and thus the friction cost method might be more appropriate to calculate lost productivity. According to Rice [31], human capital can overestimate indirect costs because the lost productivity due to premature loss of life can be replaced by existing unemployment.

To estimate lost productivity, Fenoglio et al. [32] used a human capital approach. In their research, Petrie et al. [33] argued that the willingness-to-pay approach might be better to use. The loss calculated through the willingness-to-pay approach tends to be higher compared to the loss calculated through the human capital approach. The authors Konnopka and Koenig [34] also quantified indirect costs by applying the human capital approach (with a discount rate of 5%). Using the attributable fractions, the actual conditions are underestimated. To quantify lost productivity, Chan et al. [35] and Razzouk [36] also applied the human capital approach [35,36]. In these studies, the lost productivity represents morbidity. The exact information on the wages of specific dependent patients was not known; therefore, GDP per capita was used as an approximation. The loss of productivity caused by death was calculated on the basis of the number of deaths caused by alcohol.

As stated by Chan et al., there are several differences in methods, study perspectives, jurisdictions, and cost components included in each study that yield results with great variability [35]. To compare the cost of any kind of disease or health condition, it is necessary to use more general methods or approaches that might be applied globally. Different structures of cost-of-illness studies result from the data availability and digital transformation of a country or a region. Countries/regions with higher level of digitalization are able to provide more detailed analysis with specific aims, however regions that are lagging behind in the digital transformation cannot take any advantages from the data, because these data do not exist. Razzouk [36] draws attention to the fact that many American cost-of-illness studies do not consider all types of healthcare cost; they only use the administrative data that include the general expenditure of healthcare providers covering overhead charges. There is the cost related to the treatment of specific diagnosis or the diagnostic procedures missing (such as cost of magnetic resonance, CT, X-ray, etc.).

This study aims to evaluate the economic burden of excessively drinking alcohol in the case of the Czech Republic. A similar estimation was made in the previous studies of Zabransky et al. [37] and Mlčoch et al. [38]. Mlčoch et al. [38] calculated healthcare costs separately for individual diagnoses. They also considered some diagnoses to be 100% attributable to alcohol use, and the rest of them only partially attributable. However, the attributable fractions in their studies were highly different because Mlčoch et al. [38] used attributable fractions from foreign studies. Zabransky et al. [37] based their calculation of attributable fractions on data from a registry of psychiatric care in the Czech Republic. When calculating healthcare costs, both research teams used data from one of the health insurance companies acting in the Czech Republic. They estimated the healthcare cost related to alcohol corresponding to this health insurance company, and afterwards they used an approximation to calculate the cost of all insurance companies. However, they did not take into account that people covered by other health insurance companies acting in the Czech Republic might not drink as much/as little as people covered by the company they took into consideration. Different structures of the population covered by other insurance companies might highly influence their estimation of healthcare cost related to alcohol. The structure of population covered by insurance companies might vary in many indicators, such as, and most importantly, age. However, both studies published in the Czech Republic offer different point of view on the cost estimation and are sources of attributable fractions that certainly reflect the structure of the population in Czechia. In Sweden, Ramstedt [39] estimated the healthcare cost related to alcohol by using the average daily cost and the number of days related to treatment. In quantifying the costs of unconstitutional treatment, he used data from Stockholm County, where in 2002 there were a total of 230,000 doctor visits and the cost per doctor visit was EUR 132, which was a total of EUR 30 million. Despite capital cities representing the regions which are the most developed and are different from other part of countries, their study was based on the assumption that Stockholm represents the population structure of the whole country. Healthcare cost related to alcohol is sometimes calculated in a more general way, such as that presented in the study of Fenoglio et al. [32]. They calculated deaths attributable to alcohol, hospitalizations, number of years lost, and number of days spent in hospital. They calculated the cost of institutional care by multiplying the number of hospitalizations attributable to drugs by the unit of hospitalization costs.

Ramstedt [39] emphasized that social cost related to alcohol should also be taken into account when calculating the total cost related to alcohol. He considered the institutional expenditure and expenditure on various social programs (non-institutional expenditure). To calculate the social cost attributable to drugs, he used the attributable fraction from other studies, which amounted to 19–35%. However, Ramstedt [39] did not distinguish between disability costs and incapacity costs.

To calculate the lost productivity due to mortality, Mlčoch et al. [38] and Zabransky et al. [37] distinguished between generally occurring deaths related to alcohol and deaths related to fire, traffic accidents, and injuries. They calculated the years of life indicator (YLL) and then statistically estimated the number of deaths based on the WHO attributable share of 5.3%. Mlčoch et al. [38] based their calculations on the assumption that people who had died as a result of alcohol consumption would have lived for another seven years. They calculated the lost productivity by multiplying the average annual wage by average number of years lost (seven) and by the statistically estimated number of deaths.

The international research environment uses a wide range of methods that are chosen on the basis of available databases, and also a data structure, as the presented results of the research studies indicate. Consequently, many studies have multiple methodological and data limitations. However, these limitations prevent their applicability and usage in health systems of other countries. Thus, it is very important to emphasize the creation of such studies that focus on methodological processes when evaluating the economic consequences of drinking alcohol, and subsequently share their results among other research teams in many countries in order to improve methodological processes to quantify the cost of drug use; thus, a database. The heterogeneous databases in the health systems of individual countries represent a serious barrier to the development of benchmarking indicators and in the creation of platforms for comparative studies. An absence of such studies would result in an insufficient development of preventive programs that are inevitable when dealing with a population’s addictions from various drug types. Additionally, the creation of relevant national and international policies, which focus on a decrease in the population’s addictions and a limitation of differences between countries in terms of development and a status of a particular addiction, depends on them.

This study aims to present methods and approaches that might be used to estimate the cost related to excessive alcohol consumption. It emphasizes the need for general methods and approaches that are easily applicable, because the level of digitalization and data availability vary across the regions.

The remainder of this article is organized as follows: Section 1 reviews the related theories; Section 2 briefly introduces the methods used for calculations of cost related to alcohol; in Section 3, the results are presented; Section 4 brings the study discussion; and the Section 5 presents a short conclusion.

## 2. Materials and Methods

Cost-of-illness studies vary across the world because high variabilities of data are available for their purpose. Not every country publishes data on alcohol addiction, drinking habits of its citizens, and the consequences of drinking on the whole economy. However, drinking alcohol dramatically affects the life of the population. In this study, total cost was calculated as the sum of direct healthcare cost, social care cost (concerning the expenditures on disability pensions and sickness benefits), law enforcement cost, and cost of other public authorities.

### 2.1. Data

In this study, the estimation of cost attributable to drinking was based on data from several sources such as the Institute of Health Information and Statistics of the Czech Republic (data on mortality and healthcare cost), the Czech Social Security Administration (data on sickness), and the final reports (for 2017) of different public authorities. Mortality data (or the register of deaths) contain the information on every death which occurred in 2017. It includes attributes such as age, sex, date of death, cause of death (diagnosis according to International Statistical Classification of Diseases and Related Health Problems, version 10 (ICD-10), and the place of birth/living of the person who died. 

Next, we used data (morbidity/sickness register) from the Czech Social Security Administration. The registry of sickness absence (data on morbidity) contains information of the Czech citizens; therefore, it is a register of all sick leave taken in 2017, covering working people (employees, self-employed people) or people who voluntarily pay social insurance (because only healthcare insurance is obligatory). Each case of sick leave is described by the person’s age, sex, work type, region, start and end date of sick leave, and diagnosis related to sickness. Every case of sick leave includes the information on diagnoses related to sickness; therefore, it is possible to add attributable fractions to every case.

The healthcare database contains information on how much money is spent regarding individual diagnoses. Healthcare costs are divided into hospital and ambulance costs. The data contain the sum of all costs of healthcare providers for treatment from all diagnoses connected to alcohol. Those specific diagnoses that are attributable to alcohol were identified based on a literature review performed before the evaluation of drinking costs. There are many diagnoses that are wholly attributable to drinking (i.e., alcohol is the only risk factor responsible for the outbreak of those types of diseases) or partially (i.e., there is more than one risk factor for disease outbreak). 

The costs attributable to alcohol consumption in our study were further subdivided into social cost (sickness benefit/disability pension), law enforcement, and other types of cost. To quantify the social cost attributable to alcohol, we conducted our analysis on morbidity data (registry of sickness absence) and data on disability pensions. Moreover, we also knew the average daily expenditure on sickness leave stratified by sex and age. The data on pensions are not as detailed as data on morbidity. Pension data only contain information on how many cases of disability pensions were paid by the government in 2017, regarding specific diagnoses. 

The analysis of lost productivity was based on mortality data as well as morbidity data. To estimate the law enforcement cost, we used yearly reports of the police, Administration of Justice, Ministry of Internal Affairs, and Ministry of Justice [40,41]. Furthermore, our law enforcement cost estimation was performed based on public data from the Czech police—statistics of criminality and probation and mediation service [42]. We calculated the costs to the fire brigade using data from their yearly report. We also analyzed data from the yearly report of the Finance Administration [43] of the Czech Republic. 

Sharing of data on mortality and morbidity is not applicable. However, the data on law enforcement and other types of cost data are accessible online (as cited).

For international comparison purposes, all values for 2017 were converted from Czech koruna (CZK) to USD with the purchasing power parity calculator licensed by the Massachusetts Institute of Technology [44]. Data used by the calculator come from the World Bank [45].

### 2.2. Methods

This study was based on the COI method, a descriptive method which usually does not contain any hypotheses for testing. It is used to estimate the costs of illness (or specified issue). Illness or specified issues (such as alcohol) have severe consequences in many areas of life (law enforcement, healthcare, economics, society, etc.). When applying this method, firstly, it is necessary to gain insight into those consequences and their relation to individual areas of life; therefore, knowledge of the estimation process (methods and approaches) is not sufficient. 

It determines the total financial burden of the disease (or a certain condition), taking into account direct and indirect costs. This methodology is also referred to in scientific studies as economic impact analysis (hereinafter referred to as EIA) or economic impact/disease burden (hereinafter referred to as BOD). According to the study of Møller and Matic [46], Dorothy Rice is one of the pioneers in the creation and development of COI and its use in the analysis of social costs due to alcohol use.

When calculating the cost of alcohol, it was necessary to first identify diagnoses that resulted from drinking alcohol. After that, we estimated the attributable fractions which are necessary for further calculations. Generally, authors of cost studies applied the attributable fractions based on previous studies or estimated them on their own. According to Single et al. [11], attributable fractions (AFs) are usually calculated with the formula:(1)$$\mathrm{AF}=\frac{\sum_{j=1}^{n}P_{j}\times \left(RR_{j}-1\right)}{\sum_{j=1}^{n}P_{j}\times \left(RR_{j}-1\right)+1}$$
where j is the exposure category, $RR_{\left(j\right)}$ is the relative risk at exposure level *j* compared with no consumption, and $P_{j}$ is the prevalence of the *j*th category of exposure.

As mentioned by Rockhill et al. [47], there are more than five formulas to estimate these attributable fractions. Their usage depends on data availability because the estimation of attributable fractions requires the prevalence of drinking among different groups of people. In our analysis, we used the fractions presented in Table 1 and Table 2, stated in previous studies of Zabransky et al. [37], Mlčoch et al. [38], Patra et al. [48], Collins and Lapsley [49], Shield et al. [50], Rehm et al. [51], Webster et al. [52], and Jones and Belis [53]. Both tables are a summary of the diagnoses resulting from drinking alcohol. Those diagnoses are noted according to ICD-10 (International Classification of Diseases). In Table 1, there are diagnoses wholly attributable to drinking. Therefore, their attributable fraction is 1. This fraction signifies that diagnosis results mainly from drinking alcohol.

According to previous studies of Zabransky et al. [37], Mlčoch et al. [38], Patra et al. [48], Collins and Lapsley [49], Shield et al. [50], Rehm et al. [51], Webster et al. [52], Jones and Bellis [53], diagnoses in Table 2 are only partially attributable to drinking alcohol, which means that some other risk factors exist which partially cause them. This study considers the costs related to drinking alcohol in the Czech Republic, although we mostly take into consideration the attributable fractions that were used in previous Czech studies of alcohol cost. However, only two studies exist which have dealt with this issue in this region. The first study was performed in 2011 by Zabransky et al. [37], and the second was conducted by Mlčoch et al. [38]. We conducted our own diagnoses research; therefore, there are some additional diagnoses related to alcohol that were not considered in the previous Czech studies. Table 2 contains attributable fractions mainly from Zabransky et al. [37] and Mlčoch et al. [38], but few of them were calculated as the average value of other foreign studies. Table 2 includes information on the code of diagnoses (group of diagnoses according to ICD-10), the name of diagnoses (or group of diagnoses according to ICD-10) with the corresponding attributable fractions. 

#### 2.2.1. Healthcare Cost

Zabransky [37] and Mlčoch [38] formed two research teams, which researched the estimations of social cost related to alcohol. The object of their study was also the Czech Republic. However, the availability of data and differences in research resources caused deviations in the applied methodologies/approaches. To estimate the cost of healthcare, we used the top-down approach. This approach was introduced by Bloom et al. [54] as an epidemiological approach which measures the proportion of a disease which is due to exposure to the disease or the risk factors. According to Sutton [55], the top-down approach is used to improve overall reliability and/or does not determine what the principal causes of problems may be. 

To estimate the healthcare cost, we used the attributable fractions presented in the Methods (Table 1 and Table 2). The healthcare cost was estimated from the total cost related to individual diagnoses. Healthcare cost was obtained from the Institute of Health Information and Statistics of the Czech Republic. When someone uses an ambulance because of illness (for example, related to diagnosis J00) there are some costs related to this visit (e.g., work of a doctor or a nurse). When someone is hospitalized (for example, related to diagnosis C00), there are some costs related to this hospitalization (work of a doctor, a nurse, medication, CT, magnetic resonance, etc.). Our healthcare database takes into account both ambulance and hospital costs.

We firstly identified all diagnoses attributable to alcohol and then obtained the data on cost for each diagnosis related to alcohol. For each type of healthcare (ambulance, hospital), we calculated the cost attributable to drinking alcohol according to the formula:(2)$$HC_{D,S}=C_{D,S}\times PAF_{D,S}$$
where $HC_{D,S}$ is healthcare cost which corresponds to sex and a specific diagnosis attributable to alcohol, $C_{D,S}$ is healthcare cost which corresponds to sex and specific diagnosis, and $PAF_{D,S}$ is the alcohol-attributable fraction corresponding to diagnosis and sex. 

#### 2.2.2. Social Care Cost

In our analysis, social cost consists of disability pension and sickness benefit. We calculated the sickness benefit corresponding to absences which occurred in relation to drinking alcohol. To estimate social cost, the registry of sick leaves (morbidity data) and pension data were used, as described in Section 2.1. 

The registry of sick leaves contains detailed data on every case of sick leave that occurred in 2017, as described in Section 2.1. In general, this registry covers employees and self-employed people, but also people who voluntarily pay their social insurance. We knew the diagnosis relating to specific sick leave; therefore, we joined the attributable fractions to every sick leave case in the registry of sickness absences regarding Table 1 and Table 2. We were provided with the average daily social benefit, specified by sex and region, by the Czech Social Security Administration; therefore, we calculated the sickness benefit according to the formula:(3)$$SB_{D,S}={\bar{C}}_{D,S}\times PAF_{D,S}\times l$$
where $SB_{D,S}$ is a sickness benefit that corresponds to a diagnosis attributable to alcohol, specified by sex; $C_{D,S}$ is an average daily social benefit specified by sex and region; $PAF_{D,S}$ is the alcohol-attributable fraction corresponding to diagnosis, specified by sex; and l is the period of sickness in days.

In the Czech Republic, people might be given three types of pension: old age pension, disability pension, and inheritance pension. In our analysis, we considered only disability pension, because this kind of pension only relates to drinking. To explain this in more detail, old age pension is given to every person who reaches a certain age limit (receiving the old age pension does not really depend on whether the person drank when at productive age or not). Moreover, the latter type of pension, inheritance pension, is paid to persons who have lost a family member. There is not enough evidence on whether this loss occurred as the consequence of drinking; therefore, we could not take inheritance pension into consideration. 

To estimate the cost related to disability pension, we used data provided by the Czech Social Security Administration. From these data, we determined how many disability pensions were provided in 2017, and the diagnoses to which those invalid pensions related. From the diagnosis related to disability pension, we estimated the cost related to invalid pension attributable to drinking in the same way as the social benefit cost by the following formula:(4)$$SP_{D,S}={\bar{C}}_{D,S}\times PAF_{D,S}\times m$$
where $SP_{D,S}$ is a disability pension specified by sex and diagnosis attributable to alcohol; $C_{D,S}$ is the average monthly disability pension specified by sex; $PAF_{D,S}$ is the alcohol-attributable fraction corresponding to diagnosis attributable to alcohol specified by sex; and m is the number of periods (in months).

#### 2.2.3. Law Enforcement Cost

The cost of justice was calculated based on a top-down approach; therefore, we determined the fraction which represented the amount of cost (out of total cost) related only to drinking alcohol. In the study of Zabransky et al. [37], the police cost attributable to alcohol in 2007 was estimated at USD 222.21 million [37]. The estimations of Zabransky et al. [37] were based on the psychopharmacological model and the economic model. The attributable fractions were calculated based on the statistics of committed crimes in 2007 that corresponded to the concept of the Goldstein Tripartite Conceptual framework. They assumed that crime would not be committed if there was no alcohol at all. To calculate the cost of law enforcement, Zabransky et al. [37] used the public reports which are used in the current study.

According to the Ministry of Internal Affairs [40], the total cost of policing in 2007 was USD 2370.48 million (which consists of riot and railway police services, municipal police, and traffic police service). The total cost of policing in 2017 was estimated at USD 2.38 billion. We calculated the alcohol fraction for 2017 as follows:(5)$$PAF_{A,2017}=\frac{PC_{A,2007}}{TC_{2007}}$$
where $PC_{A,2007}$ is the police cost attributable to alcohol in 2007, $TC_{2007}$ is the total cost of police in 2007, and $PAF_{A,2017}$ is the alcohol-attributable fraction of policing costs.

After that, we estimated cost attributable to alcohol in 2017 as a multiplication of the alcohol-attributable fraction ($PAF_{A,2017}$) and total cost of policing in 2017. In a similar way, we estimated the cost of courts and public prosecutor office. When calculating the cost of courts, we considered the cost of wages of judges (the number of employees is supposed to be the elementary variable, which mostly affects those types of costs). According to study of Zabransky et al. [37], in 2007 there were 2999 judges, out of which 783 dealt with criminal law. We compared the number of judges in 2007 to the number of judges in 2017. According to the Ministry of Justice [41], there were 3006 judges in 2017. To estimate the cost attributable to courts in 2017, we based our calculation on Zabransky et al. [37] and used the same attributable fraction as in their study (26.1%). We estimated the justice administration cost using the following formula: (6)$$C_{C}=PAF_{C}\times TC_{C}$$
where $C_{C}$ is the cost of courts attributable to alcohol, $TC_{C}$ is the total cost of courts, and $PAF_{C}$ is the alcohol-attributable court fraction according to Zabransky et al. [37].

The cost of public prosecutor’s office in 2017 was USD 196.94 million. This cost covers the cost related to wages of employees working in the public prosecutor’s office. According to Zabransky et al. [37], the attributable fractions relating to alcohol were 13.21%. We calculated the prosecutor’s cost attributable to alcohol as follows:(7)$$C_{P}=PP_{P}\times TC_{P}$$
where $C_{C}$ is the cost of the public prosecutor’s office attributable to alcohol, $TC_{C}$ is the total cost of the public prosecutor’s office, and $PAF_{C}$ is the alcohol-attributable fraction related to the public prosecutor’s office according to Zabransky et al. [37].

In 2017, there were 202,303 crimes committed in the Czech Republic, out of which 11,800 were committed under the influence of alcohol. Criminal activity under the influence of alcohol is usually accompanied by property damage to the owner. Individual cost of damage is estimated by experts at the place where the crime is committed. The total cost of damage incurred by criminal activity was calculated by summing the cost related to different property damages reported in the official crime statistics presented by the Police of the Czech Republic [56].

#### 2.2.4. Lost Productivity

Alcohol consumption may result in both mortality and morbidity. When death occurs, we called it a final loss of productivity. Economically, this loss represents the potential earnings (share in GDP) or time period that might be expressed, such as the potential years of life lost. In the context of drinking alcohol, it might be said that if a person did not drink, they would not die, and therefore would work for a longer period of time. 

Generally, because we do not know how long people will live, to calculate the potential years of life lost, we used the upper age limit of productivity or the life expectancy. In our analysis, we based our calculations of lost productivity (related to mortality) on life expectancy. In our analysis, we considered lost productivity due to morbidity and lost productivity due to mortality. Lost productivity due to illness was calculated based on morbidity data from the registry of sick leaves. The registry of sick leaves in the Czech Republic provides data on employees and self-employed people. In this registry, there are also people who pay their social insurance. Data on sickness absence cover the information on sick leaves regarding the diagnosis (related to sick leaves). It presents the age, sex, region, diagnosis, and start and end date of each case of absence (every person who was ill).

As it is supposed, every day of sickness presents the lost productivity. To calculate the number of days related to sickness, we subtracted the start date of sickness from the end date of sickness. We also calculated the daily GDP per capita by dividing the GDP per capita by the number of working days in a year. Finally, we estimated the lost productivity with formula below:(8)$$C_{MB}=d_{PN}\times GDP\times PP$$
where $C_{MB}$ is lost productivity attributable to alcohol due to illness (morbidity), $d_{PN}$ is the number of days of sickness, GDP is daily GDP per capita, and PP is the attributable fraction, according to the literature review presented in Section 2.2.

To estimate the lost productivity related to mortality, we used a calculation of the present value. This calculation was based on the data from the registry of mortality. Mortality data contain the information on individual deaths which occurred in the current year (2017). Every death case contains information on the age, sex, region, diagnoses related to death, etc. 

Based on the literature review, out of all deaths in the mortality register, we selected only deaths that were attributable to alcohol (according to Table 1 and Table 2). The mortality data also contain the information on how old the people were when they passed away. In general, lost productivity of people who died younger is higher than the lost productivity of people who died at an older age. To calculate the lost productivity, we used the average gross income per person, corresponding to different age levels, published by the Czech Statistical Office [57]. The human capital approach was applied. Lost productivity due to mortality was calculated as follows:(9)$$PV=\sum_{n=1}\frac{FV_{n,A}}{{\left(1+r\right)}^{T}}$$
where P*V* is the present value of future income, *FV_n,A_* is the future value of income corresponding to different age groups, *r* is the discount rate (2%), and *T* is adult life expectancy.

#### 2.2.5. Other Types of Cost

Next, we considered the cost of the Financial Administration of the Czech Republic. Previously, this type of cost was also included in an extensive study of Mlčoch et al. [38]. The Financial Administration of the Czech Republic is a public authority responsible for tax administration in the whole country. The tax system of the Czech Republic is similar in its main features to the systems of developed European countries. This institution carries out the control task of fulfilling obligations of companies as well as individuals in relation to tax administration. It detects possible tax evasion, frauds, identifies tax leakage, and ensures collection and proper administration. In relation to alcohol, there are two types of cost considered in this study: the tax leakage and the costs to the Financial Administration who provide the audits over tax administration/collection in regard to legislation.

To estimate the cost of Financial Administration related to alcohol, we only considered state expenditure on wages of employees who performed the tax audits because those costs are fixed and paid regularly. The current expenses of the Financial Administration are the costs of employees’ wages; therefore, we assumed that with the growing number of employees, these expenses might increase, and the required number of employees will increase with a growing number of inspections as well. 

According to the Ministry of Finance [58], total expenditures in 2017 amounted to USD 780.75 million. The number of employees working for the Financial Administration was 15,448. Next, the final report of Ministry of Finance [58] stated that total number of employees who performed all audits in 2017 was 3032. 

By dividing total expenses by the number of employees, we obtained the financial administration costs, which amounted to USD 50,540.57 per employee. The number of completed tax audits in 2017 was 680. This means that there are 4.46 employees per audit. From the above data on costs per employee and the average number of employees per one audit, we calculated the estimated costs of the Financial Administration of the Czech Republic attributable to alcohol. 

As stated in the final report of the Ministry of Finance [58], we calculated that for one inspection, the cost is USD 0.22 million per audit, on average. The number of controls related to alcohol was 55; therefore, we estimated the total cost to the Financial Administration attributable to alcohol was USD 12.4 million.

#### 2.2.6. Traffic Accident Cost

In 2017, 103,821 accidents occurred in the Czech Republic. According to the statistics of traffic accidents of the Czech Police presented by the Police of the Czech Republic [59], 4251 traffic accidents were caused under the influence of alcohol, out of which a driver of a motor vehicle was under the influence of alcohol (3463), the driver of a non-motorized vehicle was under the influence of alcohol (646), or a pedestrian was under the influence of alcohol (142). 

Out of all accidents caused by the drunk driver of a motor vehicle, 48 accidents resulted in the death of persons (48 killed persons), which represents 9.56% of the total number of people killed in traffic accidents (Ministry of Internal Affairs, 2018). There were 183 traffic accidents which resulted in serious injuries (203 seriously injured persons), 1441 minor injuries (1750 slightly injured persons), and 2579 accidents with material damage, presented by the Police of the Czech Republic [57].
(10)$$TD_{A}=D_{D}\times P_{D}+D_{SI}\times P_{SI}+D_{MI}\times P_{MI}+D_{MD}\times P_{MD}$$
where $TD_{A}$ is the cost of damage incurred as the consequence of traffic accidents attributable to alcohol (in USD), $D_{D}$ is the average cost of damage incurred in traffic accidents with death, $D_{SI}$ is the average cost of damage incurred in traffic accidents with serious injury, $D_{MI}$ is the average cost of damage incurred in traffic accidents with minor injury, $D_{MD}$ is the average cost of damage incurred in traffic accidents with material damage, $P_{D}$ is the number of traffic accidents with death, $PP_{SI}$ is the number of traffic accidents with serious injury, $PP_{MI}$ is the number of traffic accidents with minor injury, and $PP_{MD}$ is the number of traffic accidents with material damage.

The Fire and Rescue Service of the Czech Republic recorded 113,420 interventions in 2017. Compared to 2016, there was an increase (in 2016, there were 99,825 interventions). In 2017, there were a total of 16,757 fires in the Czech Republic, with total cost of damage estimated at USD 255.78 million. A total of 92 people died in the fires (57 of them in direct connection). On average, 46 fires occurred in the Czech Republic, while the cost of damage per day amounted to approximately USD 0.7 million. A total of 1392 people were injured in the fires. According to crime statistics presented by Police of the Czech Republic [56], 782 fires were registered as criminal activity, out of which 16 fires were committed under the influence of alcohol. According to the Police of the Czech Republic, the damage attributable to alcohol amounted to USD 2.56 million. 

To estimate costs of the Fire and Rescue Service related to alcohol, we based our estimation on the share of damage attributable to alcohol from the total calculated damage, which was calculated from the formula as follows:(11)$$PAF_{F}=\frac{D_{A}}{D_{F}}$$
where $PAF_{A}$ is the fire fraction attributable to alcohol, $D_{F}$ is the total cost of fire damage incurred (in USD-PPP (purchasing power parity)), and $D_{A}$ is the fire damage cost attributable to alcohol (in USD-PPP).

To estimate the total costs of fire and rescue service that were related to alcohol, we used the following formula: (12)$$C_{F,A}=TC_{F}\times PAF_{F}$$
where $C_{F,A}$ is the cost of the fire brigade attributable to alcohol, $PAF_{A}$ is the fire fraction attributable to alcohol, and $TC_{F}$ is the total cost of fire and rescue services (total wages paid by state).

#### 2.2.7. Cost of Social Care 

As mentioned by the Czech Social Security Administration [60], there are several types of social services provided in the Czech Republic. Their goal is to preserve the human dignity of clients, based on the individually determined needs of clients and to actively develop clients’ abilities to improve, or at least maintain, client self-sufficiency. They also provide services of the interest of clients. Providers of social services are municipalities and regions, non-governmental and non-profit organizations, and the Ministry of Labour and Social Affairs. 

Social services include social counselling, personal assistance, nursing care, emergency care, guide and reading services, support for independent living, relief services, day care centers, day and weekly hospitals, homes for the disabled, homes for the elderly, homes with a special regime, sheltered housing, and social services provided in health facilities. There are also prevention services, such as early care, telephone crisis assistance, interpreting services, shelters, halfway houses, contact centers, crisis assistance, intervention centers, low-threshold facilities for children and youth, dormitories, aftercare services, social activation services for families with children, social activation services for the elderly and people with disabilities, social therapy workshops, therapeutic communities, field programs, and social rehabilitation.

Not all types of social services or prevention programs are for clients with any kind of addiction. The costs attributable to alcohol were determined by the share of clients addicted to alcohol in individual social care providers. To calculate social costs, we conducted a questionnaire survey in the social field in cooperation with the Ministry of Labour and Social Affairs. As part of this survey, we contacted several facilities which operate various forms of social services. Approximately 50 facilities participated in this survey. Based on a questionnaire survey, it was crucial to find out what percentage of clients in each type of social services were dependent on alcohol, tobacco, illegal drugs, or gambling.

Subsequently, we used these shares in the calculation of social costs attributable to the researched issue (alcohol). We estimated these fractions as the sum of clients treated for alcohol dependence divided by the number of all clients in the facility. The resulting attributable fractions which we calculated are shown in Table 3. In the case of aftercare services, we have calculated that the proportion of clients with addiction to alcohol is 0.75. Next, we estimated the costs of social care that are attributable to alcohol by multiplying the total state expenses by the corresponding attributable fractions.

## 3. Results

As presented in Table 4, the healthcare costs attributable to alcohol were estimated at USD 527.21 million. The hospital cost was estimated at USD 464.05 million, out of which 67.24% of costs were attributable to men and the rest to women. The total ambulance healthcare cost attributable to alcohol was estimated at USD 63.15 million, 48.76% to men and 51.24% to women. There is evidence of higher healthcare costs attributable to alcohol in the case of men compared to women. To confirm this significance statistically, we provided the Wilcoxon rank test (because the normality assumption was not met). The *p*-value corresponding to hospital cost was 0.02382; therefore, we can confirm that there is a statistically significant difference in healthcare cost related to hospitalization. However, when considering ambulance cost, the evidence of higher healthcare cost in the case of men was not confirmed (corresponding *p*-value was 0.9461).

Many diagnoses are wholly attributable to alcohol, but still there are many diagnoses that are partially related to alcohol. In Table 5, the percentages of healthcare cost specified by different chapters of ICD-10 are presented. The highest cost related to alcohol was estimated for diseases of the circulatory system. The care of patients with circulatory system conditions related to alcohol presented 61.03% of all healthcare cost. Patients with neoplasms presented 11.6% of all healthcare costs related to alcohol. On the contrary, symptoms, signs, and abnormal clinical and laboratory findings, not classified elsewhere, presented only 0.03% of all healthcare costs related to drinking.

According to Table 6, total social cost attributable to alcohol was estimated at USD 0.1 billion, out of which 78.16% was related to disability pension and the rest present as the sickness benefit. 

In 2017, sickness benefit provided in relation to diagnoses that were wholly or partially related to alcohol amounted to USD 22.19 million. To explore provided sickness benefit in more detail, Table 7 incorporates insights into this type of cost (in percentage) according to ICD-10 chapters. As shown, out of USD 22.19 million, 31.46% of the provided sickness benefit in regard to alcohol was related to neoplasms. More than 30% of sickness benefit was related to diseases of the circulatory system, and 16.46% of this cost was related to mental and behavioral disorders. We found that diagnoses belonging to Y00–Y09 chapters of ICD–10 have the lowest share on provided sickness benefit due to drinking alcohol (less than 0.01%).

Total disability pension provided in relation to drinking alcohol amounted to USD 79.43 million. As stated in Table 8, the highest percentage of disability pension given was related to neoplasms (29.28%), followed by mental and behavioral disorders with 19.84%, and diseases of the digestive system (16.14%).

Drinking alcohol has several negative consequences, which can often be the cause of crime, or crime may be committed under the influence of alcohol. Crime committed under the influence of alcohol affects the amount of work of several public authorities, such as the police, prosecutor, or court. Criminal activity is usually accompanied by property damage or the other types of damage. As shown in Table 9, we estimated police cost related to alcohol at USD 223.38 million, prosecutor’s office cost at USD 26.02 million, and court costs at USD 34.12 million. The cost of damage incurred within criminal activity related to alcohol was estimated at USD 4.7 million. It can be seen that the largest cost is related to the police, and the smallest cost is related to damage incurred during criminal activity (1.63% out of USD 288.21 million).

As shown in Table 10, the lost productivity we estimated is related to both mortality and morbidity. The loss related to mortality was higher (USD 990.7 million) compared to morbidity (USD 272.31 million).

During the last few decades, alcohol smuggling and its illegal production have led to several losses in relation to alcohol. According to the Ministry of Finance (2018), significant alcohol-related leakage amounted to USD 2.22 million. As shown in Table 11, the higher leakage is connected to pure alcohol (94.95% out of total leakage related to all alcoholic drinks), followed by spirits (4.87%), beer (0.13%) and the other types of alcoholic drinks (still wine, fruit distillate, sparkling wine). All the costs in Table 11 are supposed to be the result of illegal activity/crime. Officially, it is not the tax or duty revenue. This is the money that should be paid to state, but it is not. There is also cost related to audits made within the Financial Administration of the Czech Republic (USD 12.40 million). We estimated the total cost of Financial Administration at USD 14.6 million.

In 2017, there were 48 deaths corresponding to 48 traffic accidents related to alcohol (Table 12). More than 180 accidents had 203 severe injuries, and 1441 accidents had 1850 slight injuries. The total damage incurred was an amount of USD 20.07 million. This damage represents costs mainly for insurance companies in cases where the accident was preceded by the conclusion of insurance. The estimation of damage uses the average cost of traffic accidents that was calculated by police officers in the place of accidents.

Fire service cost includes two types of cost: fire damage attributable to alcohol and personal cost (wages) of firefighters and rescue workers. Firefighters and rescue workers go beyond their limits even if the fire was started in relation to alcohol consumption; we managed to add this kind of cost to our estimation. The fire brigades are public; therefore, they are paid within state budgets, thus the costs related to firefighters and rescue workers present the cost to state, and therefore the cost to taxpayers.

We also considered fire damage to be highly related to alcohol, if the person who started the fire was accused of fire starting under the influence of alcohol. This damage was in general material, mainly property damage to its owner. It was estimated by inspectors of the accident.

In Table 13, the estimated cost of fire damage and fire/rescue brigade, based on data from Fire and Rescue Service [60], is presented. As shown, the cost related to fire damage is lower than the total personal cost of fire and rescue services attributable to alcohol (28.06%–71.94%). The total cost of fire damage and personal cost of brigades attributable to alcohol was estimated at USD 9.12 million.

According to the Czech Social Security Administration: Prague [61], there are many social services provided in the Czech Republic, however not all of them are provided to clients addicted to alcohol. There are a variety of social problems which might result in a need of specialized social care. Some types of social care services might be sought not only by addicts, but also by persons who have been indirectly affected by the addiction (such as family relatives, friends, etc.). As presented in Table 14, the highest costs related to alcohol were estimated in homes with special regimes (88.58% out of all social service cost attributable to alcohol). On the contrary, the lowest cost was estimated in the case of contact centers (0.3%).

According to Table 15, total cost related to alcohol amounted to USD 2.32 billion, which represented 0.66% of the Czech Republic GDP in 2017. We estimated that the greatest amount of alcohol cost resulted from lost productivity (54.45%); the second highest costs related to alcohol were estimated in healthcare, at nearly 30%. On the contrary, the lowest cost related to alcohol corresponded to the fire and rescue brigade (0.39%). 

## 4. Discussion

This study collates insight into methods and approaches that are focused on estimations of the social cost of detrimental alcohol consumption. It emphasizes the need for more generalizable methods, because there are a variety of social cost-of-illness studies that differ in several aspects (data used, different cost structures, different methods, etc.). Moreover, it presents a real estimation of the cost of alcohol in the case of the Czech Republic in 2017. It shows an appropriate generalized estimation of social cost related to alcohol consumption and describes the data used for this estimation in detail. The methods, approaches, data, and structure of cost-of-illness differ; therefore, the results of the studies already published are not comparable, and thus it does not allow researchers to benchmark countries (regions) regarding the cost of alcohol.

In this study, we estimated the cost related to alcohol at USD 2.32 billion, which represents 0.66% of Czech GDP. The highest cost related to alcohol resulted from lost productivity (54.45% of total cost, USD 1.2 billion). This type of cost is considered to be indirect, because it is not directly attributable to alcohol. We distinguished between lost productivity attributable to mortality and morbidity. Lost productivity related to mortality represented the sum of productivity losses in all age groups and genders. Considering the hybrid prevalence and incidence approach in the estimation of lost productivity was the subject of a study by Sorge et al. [62]. The lost productivity attributable to morbidity was lower than the lost productivity attributable to mortality, which might arise from the fact that lost productivity of morbidity in our study was only temporary. 

The cost related to lifetime morbidity was included in the estimation of social care cost—disability pensions. Working performance of people with higher levels of disability might be lower, therefore their work abilities are limited to physically or psychically easier positions, with specific working times (e.g., part-time job). People with the highest degree of invalidity are not expected to work at all, therefore we did not include “them” in calculated lost productivity. We assumed that people with disability are not supposed to present lifetime lost productivity, because their work position might be replaced by someone else. However, the calculation of friction cost would be helpful. 

In our study, we considered healthcare costs, social care and service costs, law enforcement costs, fire service costs, cost of damage incurred during traffic accidents, and other public administration costs. The attributable fractions related to mortality were applied in the calculation of healthcare costs, social care costs, and lost productivity. To estimate the lost productivity related to mortality, we applied a human capital approach. A top-down approach was applied to estimate healthcare cost, social care cost, and law enforcement. A bottom-up approach was applied in calculations of Public Administration costs and traffic accident costs.

Before conducting this study, we aimed to apply a friction cost method and willingness-to-pay approach, because both are also considered to be relevant for this kind of study. However, we were limited by the data. Our databases limited us even in calculations of the DALY indicator, which is generally used to quantify lost productivity and the consequences of disease or disability. We also considered estimations of prison cost, but the Czech Republic only provide aggregated data without regard to alcohol specifically. The estimation of lost productivity related to imprisonment was also considered. 

As mentioned in the Introduction, there are harmful effects related not only to the users of alcohol, but also to their social network, community, and families. Addiction may result in debts and associated existential problems. People addicted to alcohol or other types of drug might commit crimes to gain money. The consequence of drinking might be death. If person who takes care of other family members becomes addicted, the family loses income. Furthermore, if the “breadwinner” of the family dies as a result of alcohol addiction, it might have devastating effects on all family members. Even though there are many harmful effects of excessive drinking, it is not easy to quantify all alcohol consumption consequences mentioned above.

In the previous study of Zabransky et al. [37], the authors conducted an analysis of social cost related to alcohol in 2011, estimating this cost at USD 1.15 billion (in PPP), which represented 0.46% of GDP. Mlčoch et al. [38] estimated the cost of alcohol at USD 3.96 billion, which represented 1.2% of national GDP in 2016. According to Mlčoch et al. [38], the share of lost productivity in GDP was 41.4% (USD 1.7 billion in PPP). The lost productivity estimation in our study was based on real data from the registry of sick leaves in the Czech Republic. On the contrary, Mlčoch et al. [38] based their analysis only on estimations of absenteeism and presenteeism which are mainly theoretical concepts; thus, we consider our estimation to be more precise. To calculate the lost productivity related to mortality, we applied a human capital approach. Mlčoch et al. [38] calculated lost productivity related to mortality based on statistics from Germany. They performed only an approximation of general mortality to the whole population. 

While we performed an analysis of healthcare costs based on data from the Institute of Health Information and Statistics, which aggregates the data from all health insurance companies, Mlčoch et al. [38] used data only from one insurance company, calculated its market share, and then performed an approximation. As a result, we consider our estimation of healthcare cost to be more exact. Furthermore, in the research of Varney and Guest [63], the authors used attributable fractions based on relative risks of individual diagnoses from out-sourced data. Fractions were calculated based on the number of visits of general practitioners. The proportion of visits attributable to alcohol was calculated for each level of alcohol consumption. They also included the cost of hospitalizations in their estimation of cost. However, to estimate the ambulance cost related to alcohol, they used fractions calculated in previous studies. We consider the approach of Varney and Guest to be predisposed to overestimates of healthcare costs.

To estimate law enforcement cost related to alcohol, the attributable fractions from Zabransky et al. [37] were applied. The same approach was applied by Ramstedt [39], who estimated the cost of policing related to drug abuse based on an annual report of Swedish police and the assumption that 30% of police costs in Sweden are related to illegal drug use; thus, we consider this approach to be important and appropriate for this kind of calculation. Different procedures for quantifying law enforcement costs have been chosen by Zabransky et al. [37] and Mlčoch et al. [38]. 

In our study, we considered generalization of the methodology, therefore the approach applied by Mlčoch et al., who used secondary Eurostat data to quantify law enforcement costs (courts and police), could also be assumed. To estimate the law enforcement cost related to alcohol, Mlčoch et al. [38] used an attributable fraction, calculated from criminal statistics. Mlčoch et al. [38] divided the number of crimes attributable to alcohol by the total number of crimes committed in the Czech Republic in 2016. This attributable fraction was calculated to 5.52%. The cost attributable to alcohol was then calculated as the multiplication of the public order and security expenditure according to the COFOG classification and the calculated attributable fraction (5.52%). We wanted to further extend our calculation; therefore, we used data from the official report of the Ministry of Interior. Moreover, we wanted to perform the calculations of cost related to courts and police separately. In the study of Mlčoch et al. [38], the authors had incorrectly calculated the cost related to police and to courts, because they based their calculations on public order and security expenditure. 

The relationship between excessive drinking of alcohol and mortality might differ from the relationship between excessive drinking of alcohol and morbidity. It arises because drinking does not immediately cause death. Excessive drinking increases the risk of health issues. Considering only internal causes of death, if health issues are not treated for a longer period of time, the probability of dying increases. However, if people addicted to alcohol start to recover from addiction, their health might improve, and so the death would not occur at all. Our estimation included the attributable fractions calculated from mortality; thus, this could underestimate our estimated cost. The calculation of attributable fractions from the data of morbidity might take place in our future studies. The attributable fractions calculated from morbidity data could be applied to estimate the cost related to social care. Positive effects of alcohol consumption are now considered to be controversial. This controversy results from a heterogeneity of views of authors on limits for the “safe” consumption of alcohol and the negative effects of consumption that overcome the positive effects. Chikritzhs et al. [64] do not recommend assumptions of any protective effects from low dose consumption. Next, the authors stated that health benefits from moderate drinking should no longer play a role in decision making. Furthermore, Sherk et al. [65] suggested a need of downward revisions to daily consumption levels, which would result in safe drinking with no added risks. On the contrary, many authors, such as Sacks et al. [66], have considered the positive effects of drinking.

Our study could be further extended, and this extension might concern the calculation of actual attributable fractions of diagnoses related to alcohol. The implementation of alternative procedures depends on the data availability and data sources; therefore, the time spent by data collection might be estimated as period of more than three years. However, this may encounter legislative problems. Legislative problems of data processing can be solved, for example, by security checks or other tools by which the data of citizens of the Czech Republic can be handled. Extension might concern three different changes: change of attributable fractions; change of data provider; and change of used data. Those changes would be highly influenced by data availability. Data availability makes the appropriateness of some methodologies more optimal than others. Change of data provider would make it possible to use the same methods as those used in the current study. At the same time, such a change may also represent a change in the data provided, because data providers may provide data in a different structure, for example. The combination of all three proposed changes might be taken into consideration as well. The recalculation of the attributable fraction will affect the cost in all different fields mentioned in our analysis. The extension might also be conducted in regard to geographical locations. While our study included estimations of cost regarding the Czech Republic, this kind of study might also include the estimation of cost attributable to alcohol in regard to several regions. An excellent example concerning the cost across several geographical points is the study of Miller et al. [67], who estimated the cost related to cities in the United States. In order to reduce the costs associated with alcohol consumption, Orosová et al. [68], Horáková et al. [69], and Rolová [70] recommend improving prevention, which may include an increase in prices of alcoholic beverages (or at least increases in excise duties or even a partial ban on the consumption and sale of alcoholic beverages). In such a case, however, it is also necessary to consider that people who are dependent on alcohol may be at greater risk of criminal activity in order to obtain money to buy alcoholic beverages. Regarding alcohol addicts, the risk of crimes committed under the influence of alcohol or suicide attempts might become higher because the alcoholic restrictions would be more significant.

## 5. Conclusions

Lost productivity and avoidable mortality and morbidity are detrimental consequences of drinking. To estimate the cost related to alcohol, the cost-of-illness method was used. However, many cost-of-illness studies significantly differ from each other. The variability of those studies results from different cost structures, data availability, and methods/approaches applied. When considering all the differences, it is almost impossible to perform international benchmarking of countries regarding the costs related to specific issues (such as alcohol drinking). The benchmarking of countries in regard to costs resulting from several health conditions would help international authorities to create policies towards excessive drinking in a broader way. To reduce the cost related to alcohol and other specific issues, it is important to adopt legal procedures on reducing harmful effects of legal drugs. In general, to reduce demand, it is necessary to increase the price of a commodity. In the case of legal drugs, it is possible to increase the price by increasing the excise tax. It is also necessary to educate people on the harmful effects of drinking. 

This study applied the methods and approaches to estimate costs related to alcohol that are easy to be explained, applied, and understood. Our analysis used an estimation of cost related to alcohol in healthcare, social care and services, law enforcement, fire and rescue services, Public Administration, and traffic accidents. Generally, the application of methods depends on the available database and the main aim of the study. On the contrary, this study was mainly based on the data that are open-sourced and might be used in further cost-of-illness studies.

In this study, total cost of alcohol was estimated at USD 2.32 billion, which is 0.66% of Czech GDP. Lost productivity represented the highest cost related to alcohol. The lost productivity related to mortality was higher than lost productivity related to morbidity. Lost productivity, especially in cases of morbidity, is usually related to healthcare cost. Therefore, the costs of healthcare were the second highest in our estimation. However, there are many other things influenced by alcohol drinking. When people are under the influence of alcohol, they might cause a car accident, or commit a crime. People influenced by alcohol represent a danger to their surroundings and to themselves as well. 

There are many methods to estimate the cost of alcohol (or other health conditions and environmental issues). One of them is to consider the attributable fraction estimated not only from mortality data, but from morbidity data as well. Morbidity attributable fractions would mostly change the cost related to social care. The estimation of cost related to law enforcement might be performed according to specific statistical approaches. However, the specificity of applied methods makes it impossible to other research teams to apply the same procedures. 

This study’s results represent an available platform for the creators of strategic health plans and policymakers as well. Regarding the results, policymakers will be able to identify “where” alcohol consequences arise the most, and how much the alcohol consequences cost. Additionally, from a methodological and research point of view, these results will support the development of important benchmarking indicators in this area, creating a significant platform for the creation of comparative studies between those countries which are absent now. Methods and approaches applied in our study might be useful for research teams who cooperate in policymaking. If researchers use the same methods and approaches over longer period (more years), it will be possible to compare the results and to decide whether applied restrictions (regarding alcohol) impact the cost of alcohol. 

The creation of active preventive programs and effective policies that focus on the elimination of a population’s alcohol consumption may be supported by their availability. At present, this is an ambition of each country, and is also a part of the strategic goals of national and international health institutions. Active and successful drug policy will be evident in positive impacts on society, and also the countries’ economies. The financial sustainability of national health and social systems, which has constantly been proclaimed during the last decades, would also be impacted by this policy. 

## Figures and Tables

**Table 1 ijerph-18-04964-t001:** Diagnoses wholly attributable to alcohol.

ICD-10 Code	ICD-10 Diagnosis Name	AF	References
E24.4	Alcohol-induced pseudo-Cushing syndrome	1	by def., [38,51,53]
F10	Mental and behavioral disorders due to use of alcohol	1	by def., [37,38,48,49,50,51,52,53]
G31.2	Degeneration of nervous system due to alcohol	1	by def., [38,48,50,51,52,53]
G62.1	Alcoholic polyneuropathy	1	by def., [38,48,49,50,51,52,53]
G72.1	Alcoholic myopathy	1	by def., [38,50,51,52,53]
I42.6	Alcoholic cardiomyopathy	1	by def., [38,48,50,51,52,53]
K29.2	Alcoholic gastritis	1	by def., [48,49,50,51,52,53]
K70	Alcoholic liver disease	1	by def., [38,49,50,51,52,53]
K73	Chronic hepatitis, not classified elsewhere	1	by def., [38,52]
K85.2	Alcohol-induced acute pancreatitis	1	by def., [38,50,51,52]
K86.0	Alcohol-induced chronic pancreatitis	1	by def., [48,50,51,52,53]
O35.4	Maternal care for (suspected) damage to fetus from alcohol	1	by def., [38,48,51,52]
P04.3	Fetus and new-born affected by maternal use of alcohol	1	by def., [38,50,51]
Q86.0	Fetal alcohol syndrome (dysmorphic)	1	by def., [38,50,51,53]
R78.0	Finding of alcohol in blood	1	by def., [38,48,51,52,53]
T51	Toxic effect of alcohol	1	by def., [51,52]
X65	Intentional self-poisoning by and exposure to alcohol	1	by def., [49,51,52]
Y15	Poisoning by and exposure to alcohol, undetermined intent	1	by def., [48,51,52]
Y90	Evidence of alcohol involvement determined by blood alcohol level	1	by def., [51,52,53]
X45	Accidental poisoning by and exposure to alcohol	1	by def., [48,52]

Source: by def., Zabransky et al. [37], Mlčoch et al. [38], Patra et al. [48], Collins and Lapsley [49], Shield et al. [50], Rehm et al. [51], Webster et al. [52], and Jones and Belis [53]. AF, attributable fraction; ICD-10, International Statistical Classification of Diseases and Related Health Problems, version 10.

**Table 2 ijerph-18-04964-t002:** Diagnoses partially attributable to alcohol.

ICD-10 Code	ICD-10 Name	AF Women	AF Men	References
A15–A19	Tuberculosis	0.12	0.209	[51,52,53]
C00–C06	Malignant neoplasms of lip, tongue, unspecified parts of tongue, gum, floor of mouth, palate, other and unspecified parts of mouth	0.176	0.614	[37,38,48,49,50,51,52,53]
C07,C08	Malignant neoplasms of parotid gland and other unspecified major salivary glands	0.176	0.614	[37,48,50,51,52,53]
C09,C10	Malignant neoplasms of tonsil, oropharynx	0.176	0.614	[37,38,48,49,50,51,52,53]
C11	Malignant neoplasms of nasopharynx	0.176	0.614	[37,48,50,53]
C12–C14	Malignant neoplasms pirifrom sinus, hypopharynx and of other and ill-defined sites in the lip, oral cavity, and pharynx	0.176	0.614	[37,38,48,49,50,51,52,53]
C15	Malignant neoplasm of esophagus	0.197	0.591	[37,38,48,49,50,51,52,53]
C16	Malignant neoplasm of stomach	0.068	0.192	[37]
C18	Malignant neoplasm of colon	0.17	0.17	[38,50,52,53]
C19	Malignant neoplasm of rectosimoid junction	0.17	0.17	[38,50,52]
C20	Malignant neoplasm of rectum	0.17	0.17	[38,50,52,53]
C21	Malignant neoplasm of anus and anal canal	-	0.17	[38]
C22	Malignant neoplasm of liver and intrahepatic bile ducts	0.145	0.538	[37,38,48,49,50,51,52,53]
C25	Malignant neoplasm of pancreas	0.013	0.022	[52]
C32	Malignant neoplasm of larynx	0.221	0.655	[37,38,48,49,50,51,52,53]
C50	Malignant neoplasm of breast	0.15	-	[37,38,48,49,50,52,53]
C61	Malignant neoplasm of prostate	0.17	-	[38]
D00–D09	In situ neoplasms	0.037	0.199	[37,48,50]
D10–D36	Benign neoplasms	0.037	0.199	[37,48,50]
D37–D48	Neoplasms of uncertain or unknown behavior	0.037	0.199	[37,48,50]
G40	Epilepsy	0.238	0.747	[37,48,49,50,53]
G41	Status epilepticus	0.238	0.747	[37,48,49,50,53]
I10–I15	Hypertensive diseases	0.29	0.3	[37,38,48,49,52,53]
I47	Paroxysmal tachycardia	0.114	0.372	[37,38,48,49,51,52,53]
I48	Atrial fibrillation and flutter	0.114	0.372	[37,38,48,49,50,52,53]
I49	Other cardiac arrhythmias	0.114	0.372	[37,48,51,53]
I60–I62	Subarachnoid hemorrhage, intracerebral hemorrhage, other nontraumatic intracranial hemorrhage	-	0.05	[37,48,49,51,52,53]
I63–I66	Cerebral infarction, stroke, not specified as hemorrhage or infarction, occlusion and stenosis of precerebral arteries, not resulting in cerebral infarction, occlusion and stenosis of cerebral arteries, not resulting in cerebral infarction	-	0.174	[37,48,49,50,51,52,53]
I67–I69	Other cerebrovascular diseases, cerebrovascular disorders in diseases classified elsewhere, sequelae of cerebrovascular disease	0.015	0.056	[37,48,49]
I85	Esophageal varices	0.277	0.821	[37,48,49,53]
J10–J18	Influenza and pneumonia	0.031	0.062	[51,52]
K22.6	Gastro-esophageal laceration–hemorrhage syndrome	0.41	0.41	[49]
K74.3	Primary biliary cirrhosis	0.57	0.57	[38,48,51]
K74.4	Secondary biliary cirrhosis	0.57	0.57	[38,48,51]
K74.5	Biliary cirrhosis, unspecified	0.4	0.4	[38,48,51]
K74.6	Other and unspecified cirrhosis of liver	0.4	0.4	[38,48,51]
K76.6	Portal hypertension	0.4	0.4	[38,48,51]
L40	Psoriasis	0.351	0.351	[37,38,48]
O03	Spontaneous abortion	0.015	-	[37,38,51]
P05	Slow fetal growth and fetal malnutrition	0.017	-	[37,48,51]
R96–R99	Other sudden death, cause unknown, unattended death, other ill-defined and unspecified causes of mortality	0.1	0.1	[52]
W00–W19	Falls	0.047	0.047	[48,49,52,53]
X00–X09	Exposure to smoke, fire, and flames	0.041	0.041	[48,49,52,53]
X41	Accidental poisoning by and exposure to antiepileptic, sedative-hypnotic, anti-parkinsonism, and psychotropic drugs, not elsewhere classified	0.154	0.154	[48,52,53]
X42	Accidental poisoning by and exposure to narcotics and psychodysleptics (hallucinogens), not classified elsewhere	0.154	0.154	[48,52,53]
V01–V99	Transport accidents	0.041	0.03	[37,49,52]
W20–X59	Other external causes of accidental injury	0.03	0.03	[48,52]
X60–X84	Intentional self-harm	0.153	0.041	[37,48,49,52,53]
X85–Y09	Assault	0.041	0.041	[48,49,52]
Y10–Y34	Event of undetermined intent	0.041	0.041	[52]
Y40–Y84	Complications of medical and surgical care	0.041	0.041	[52]
Y85–Y89	Sequelae of external causes of morbidity and mortality	0.041	0.041	[52]
Y90–Y98	Supplementary factors related to causes of morbidity and mortality classified elsewhere	0.041	0.041	[52]

Source: Zabransky et al. [37], Mlčoch et al. [38], Patra et al. [48], Collins and Lapsley [49], Shield et al. [50], Rehm et al. [51], Webster et al. [52], Jones and Bellis [53].

**Table 3 ijerph-18-04964-t003:** Attributable fractions corresponding to different type of social services.

Type of Social Services	Attributable Fraction
Aftercare service	0.75
Therapeutic communities	0.22
Home with special regime	0.86
Professional social counselling	0.28
Contact center	0.07
Field programs	0.05

**Table 4 ijerph-18-04964-t004:** Healthcare cost attributable to alcohol in million USD, 2017.

Healthcare Type	Cost (Million USD-PPP)	Men (%)	Women (%)
Hospital	464.05	67.24	32.76
Ambulance	63.15	48.76	51.24
Total cost (million USD)	527.21		

PPP, purchasing power parity.

**Table 5 ijerph-18-04964-t005:** Healthcare cost stratified by type of disease, attributable to alcohol in percentage, 2017.

ICD-10 Code	ICD-10 Chapter	Cost (Thousand USD-PPP)	Cost (%)
A00–B99	Certain infectious and parasitic diseases	263.6	0.05
C00–D48	Neoplasms	61,177.06	11.60
E00–E90	Endocrine, nutritional, and metabolic diseases	-	-
F00–F99	Mental and behavioral disorders	34,711.29	6.58
G00–G99	Diseases of the nervous system	33,762.32	6.40
I00–I99	Diseases of the circulatory system	321,775.33	61.03
J00–J99	Diseases of the respiratory system	9985.3	1.89
K00–K93	Diseases of the digestive system	34,184.08	6.48
L00–L99	Diseases of the skin and subcutaneous tissue	8034.63	1.52
O00–O99	Pregnancy, childbirth, and the puerperium	1739.78	0.33
P00–P96	Certain conditions originating in the perinatal period	7707.76	1.46
Q00–Q99	Congenital malformations, deformations, and chromosomal abnormalities	-	-
R00–R99	Symptoms, signs, and abnormal clinical and laboratory findings, not elsewhere classified	158.16	0.03
S00–T98	Injury, poisoning and certain other consequences of external causes	843.53	0.16
V01–X59	Accidents	8804.35	1.67
X60–X84	Intentional self-harm	210.88	0.04
X85–Y09	Assault	316.32	0.06
Y10–Y34	Event of undetermined intent	263.6	0.05
Y40–Y84	Complications of medical and surgical care	3005.08	0.57
Y85–Y89	Sequelae of external causes of morbidity and mortality	-	-
Y90–Y98	Supplementary factors related to causes of morbidity and mortality classified elsewhere	263.6	0.05
Total healthcare cost		527,206.67	100

**Table 6 ijerph-18-04964-t006:** Social care cost attributable to alcohol in million USD and percentage, 2017.

Cost Type	Cost (Million USD-PPP)	Share (%)		
		I Type *	II Type **	III Type ***
Disability pension	79.43	22.34	15.12	62.53
Sickness benefit	22.19	-		
Total cost (million USD)	101.63			

* Decline in working capacity by 35–49%, ** decline in working capacity by 50–69%, *** decline in working capacity by 70% or more.

**Table 7 ijerph-18-04964-t007:** Social care cost attributable to alcohol—sickness benefit in percentage, 2017.

ICD-10 Code	ICD-10 Chapter	Cost (Thousand USD-PPP)	Cost (%)
A00–B99	Certain infectious and parasitic diseases	102.09	0.46
C00–D48	Neoplasms	6981.89	31.46
F00–F99	Mental and behavioral disorders	3652.95	16.46
G00–G99	Diseases of the nervous system	1600.11	7.21
I00–I99	Diseases of the circulatory system	6813.23	30.70
J00–J99	Diseases of the respiratory system	545.95	2.46
K00–K93	Diseases of the digestive system	1981.83	8.93
L00–L99	Diseases of the skin and subcutaneous tissue	481.59	2.17
O00–O99	Pregnancy, childbirth, and the puerperium	6.66	0.03
R00–R99	Symptoms, signs, and abnormal clinical and laboratory findings, not elsewhere classified	4.44	0.02
S00–T98	Injury, poisoning, and certain other consequences of external causes	6.66	0.03
V00–V99	Accidents	2.22	0.01
W00–X59	Other external causes of accidental injury	11.1	0.05
Y10–Y34	Event of undetermined intent	2.22	0.01
Y00–Y09	Assault	0.02	<0.01
Total social care cost—sickness benefit		22,192.96	100

**Table 8 ijerph-18-04964-t008:** Social care cost attributable to alcohol, disability pension in percentage, 2017.

ICD-10 Code	ICD-10 Chapter	Cost (Thousand USD-PPP)	Cost (%)
A00–B99	Certain infectious and parasitic diseases	262.13	0.33
C00–D48	Neoplasms	23,257.74	29.28
F00–F99	Mental and behavioral disorders	15,759.34	19.84
G00–G99	Diseases of the nervous system	15,171,54	19.10
I00–I99	Diseases of the circulatory system	8308.61	10.46
J00–J99	Diseases of the respiratory system	23.83	0.03
K00–K93	Diseases of the digestive system	12,820.35	16.14
L00–L99	Diseases of the skin and subcutaneous tissue	3828.63	4.82
Total social care cost—disability pension		79,432.17	100

**Table 9 ijerph-18-04964-t009:** Law enforcement cost in million USD, 2017.

Cost Type	Cost Related to Alcohol (Million USD-PPP)	Cost Related to Alcohol (%)
Police	223.38	77.50
Public prosecutor’s office	26.02	9.03
Courts	34.12	11.84
Criminal activity damage	4.7	1.63
Total cost (million USD)	288.21	-

Source: Ministry of Justice [41], Ministry of Internal Affairs [40], Police of the Czech Republic [56,59].

**Table 10 ijerph-18-04964-t010:** Lost productivity attributable to alcohol in million USD, 2017.

Lost Productivity	Morbidity	Mortality
Lost productivity (million USD) in PPP	272.31	990.7

**Table 11 ijerph-18-04964-t011:** Leakage of Excise Duty, VST, Duty related to alcohol.

Commodity	Count	Leakage ED, VAT, Duty (Thousand USD-PPP)	Leakage (%)	Value (Million USD)
Alcohol	16	2104.37	94.95	-
Spirits	485	107.87	4.87	-
Fruit distillate	2	0.25	0.01	-
Beer	27	2.89	0.13	-
Still wine	22	0.78	0.04	-
Sparkling wine	6	0.05	<0.01	-
Audits	55	-	-	12.4
Audits + leakage	-	-	-	14.6

ED, excise duty; VAT, value-added tax. Source: Ministry of Finance [58].

**Table 12 ijerph-18-04964-t012:** Traffic accident damage in million USD, 2017.

Traffic Accident	Number of Accidents	Number of Persons
With death	48	48 deaths
With severe injury	183	203 severely injured
With slight injury	1441	1850 slightly injured
With material damage	2579	-
Total accident damage in million USD	20.07	-

Source: Police of the Czech Republic [59].

**Table 13 ijerph-18-04964-t013:** Fire damage and cost of fire and rescue services attributable to alcohol, 2017.

Cost Type	Cost (Million USD-PPP)	Cost (%)
Fire damage attributable to alcohol	2.56	28.06
Personal cost of fire and rescue service attributable to alcohol	6.56	71.94
Total cost of fire and rescue brigade attributable to alcohol	9.12	100

Source: Fire and Rescue Service [60], Police of the Czech Republic [56].

**Table 14 ijerph-18-04964-t014:** Social services costs attributable to alcohol, 2017.

Social Service Type	Cost (Thousand USD-PPP)	Cost (%)
Aftercare service	2126.66	2.21
Therapeutic communities	506.34	0.53
Home with special regime	85,383.11	88.58
Professional social counselling	7449.84	7.73
Contact center	286.54	0.30
Field programs	642.23	0.67
Total cost of social care in thousand USD	96,394.73	-

**Table 15 ijerph-18-04964-t015:** Total cost related to alcohol.

Cost Type	Cost (Million USD-PPP)	Cost (%)
Healthcare	527.21	22.71
Social care	101.63	4.38
Law enforcement	288.21	12.4
Lost productivity	1263.71	54.45
Financial Administration	14.6	0.63
Traffic accidents	20.07	0.86
Fire and Rescue brigade	9.12	0.39
Social services	96.39	4.16
Total cost	2320.97	100

## Data Availability

The data that support the findings of this study are available from the corresponding author, upon reasonable request.

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
