# Peer review of "Methods for Estimating Avoidable Costs of Excessive Alcohol Consumption"

_ijerph, 2021, doi:10.3390/ijerph18094964_

Round 1
Reviewer 1 Report
The manuscript provides a social cost analysis of alcohol use in the Czech Republic with the major cost areas considered being lost work place productivity, health care and law enforcement. The authors outline the difficulties of assembling administrative data and attributing costs and the underpinning assumptions.
Introduction
- I believe that the paragraph on COVID-19 is unnecessary and can be deleted (also the opening to the abstract)
- Page 1 line 38: treatment programs are for those with alcohol use problems or disorders: prevention programs are to reduce the uptake of substance use or to prevent transition to problematic substance use.
- Page 1 line 38/39 Just clarify that alcohol-caused harms occur across the spectrum of use – including harms to non-users.
- Page 2 lines 46-86 – please split into paragraphs
- Page 2 line 52 (and elsewhere) you need to either descript the results of the study and then add the reference or put in the name of at least one author e.g. “Cost related to mental disorders was estimated by Angelis et al [3] ”
- Page 2: My understanding is that authors can limit social cost studies to any domain, so I’m not sure that this is a criticism of the social cost method. Perhaps just use this section to illustrate the diverse range of costs that have been explored?
- Page 3 105. I thought the frictional period was the time until a worker could be replaced (or is that only in cases of death). What about a worker who is temporarily replaced while they are sick.
Method
- Page 6 line 264 “Czech Social Security Administration. Sickness register”. Who does this register cover? Only those in full-time employment? Self-employed, “gig” (informal economy)? Presumably no value to voluntary work or housework? Is there anyway of valuing these other forms of lost productivity?
- Table 1 – you include Q86.0 FAS – yes caused by alcohol, but most studies report that they are unable to estimate the AF due to poor / missing data on the prevalence of alcohol use by women during pregnancy. Did you manage to find these data?
- Previous social cost of alcohol include estimates of the protective effects of alcohol for some condition. While the extent of these protective effects is increasingly contested (Chikritzhs, T., Stockwell, T., Naimi, T., Andreasson, S., Dangardt, F., Liang, W., 2015. Has the leaning tower of presumed health benefits from ‘moderate’ alcohol use finally collapsed? Addiction 110, 726-727: Sherk, A., Gilmore, W., Churchill, S., Lensvelt, E., Stockwell, T., Chikritzhs, T., 2019. Implications of cardioprotective assumptions for national drinking guidelines and alcohol harm monitoring systems. International Journal of Environmental Research and Public Health 16, 4956) there are some conditions (e.g. diabetes type 2 women only: Ischaemic heart disease) that are generally still included. I couldn’t work out how you addressed “protective effects”.
- Law enforcement: I appreciated the difficulty of developing an “AF” from these data. However, please give your reader more information on how Zabransky et al. arrived at their figure.
- What about the costs of prisons?
- (Minor point – generally contractions (e.g. didn’t) should not be used)
- Table 4 and elsewhere. Consider including an online table with your main results converted to USD or Euro for international readers – there are online purchasing power parity calculators to help with that, rather than just using the exchange rate.
- Table 11 and section 2.2.5. I thought that taxes and duties were transfer costs and weren’t included in social cost studies. I think that the audits are eligible for inclusion, but perhaps include a reference in 2.2.5 to earlier studies that have included these lost payments?
- Minor point line 621 typo “More than 180 accidents with 203 SEVERE injuries”
- Minor point line 647 Typo “Total cost related to alcoholtotal cost related to alcohol”
Discussion
- “The lost productivity attributable to morbidity is lower than the lost productivity attributable to mortality, what arises from the fact that morbidity is only temporary”. This ignores morbidity that results in long term disability or impairment (which is very hard to value)
- “The calculation of attributable fractions from the data of morbidity might take place in future studies. The attributable fractions calculated from morbidity data would be applied to estimate the cost related to social care” Separate alcohol morbidity AF are available. See appendices to Collins DJ, Lapsley HM. The costs of tobacco, alcohol and illicit drug abuse to Australian society in 2004/05. Canberra: Commonwealth of Australia; 2008.
- Line 761 “or a total ban on the consumption and sale of alcoholic beverages [61-63].” Given the failures of alcohol prohibition in the USA and the devastating effects of the “war on drugs” There may be unintended consequences and costs from this suggestion.
- Social cost studies are entitled to draw the parameters of their calculation where ever they please. However, I think that even if you don’t estimate the cost, you should at least acknowledge the extent of harms to others for alcohol use. Other than traffic crashes, I don’t think that these will appear in your costing. While not economic costing studies, alcohol is the only substance thought to cause more harm to others than to the consumer (Bonomo Y, Norman A, Biondo S, Bruno R, Daglish M, Dawe S, et al. The Australian drug harms ranking study. Journal of Psychopharmacology. 2019;33(7):759-68. Nutt DJ, King LA, Phillips LD. Drug harms in the UK: A multicriteria decision analysis. Lancet. 2010;376(9752):1558-65
-
The journal asks reviewers to comment on the adequacy of the literature reviewed – while you include many that I’m unfamiliar with, you might include additional studies from the USA (e.g. Miller TR, Nygaard P, Gaidus A, Grube JW, Ponicki WR, Lawrence BA, et al. Heterogeneous costs of alcohol and drug problems across cities and counties in California. Alcohol Clin Exp Res. 2017;41(4):758-68. Sacks JJ, Gonzales KR, Bouchery EE, Tomedi LE, Brewer RD. 2010 national and state costs of excessive alcohol consumption. Am J Prev Med. 2015;49(5):e73-e9. and Canada (e.g Sherk A, Stockwell T, Rehm J, Dorocicz J, Shield KD, Churchill S. The International Model of Alcohol Harms and Policies: A new method for estimating alcohol health harms with application to alcohol-attributable mortality in Canada. J Stud Alcohol Drugs. 2020;81(3):339-51. Sorge JT, Young M, Maloney-Hall B, Sherk A, Kent P, Zhao J, et al. Estimation of the impacts of substance use on workplace productivity: A hybrid human capital and prevalence-based approach applied to Canada. Can J Public Health. 2020:1-10.) The Sorge et al paper focuses on workplace productivity so may be of particular interest to you.
Author Response
Dear Reviewer,
Thank you very much for the careful and thorough reading of our manuscript and for the thoughtful comments and constructive suggestions. We hope that these revisions improve the paper such that you now deem it worthy of publication in „International Journal of Environmental Research and Public Health“ and our revision has improved the paper to a level of your satisfaction. We have carefully revised the paper and rewrite accordingly. In the following we respond in detail to the all comments. All changes in the revised manuscript are highlighted using the "Track Changes" function.
Thank you for giving us the opportunity to revise our manuscript. We would like to send our revised study again for considering and we look forward to hearing from you.
Kind regards,
Beata Gavurova and Miriama Tarhanicova
Open Review
Comments and Suggestions for Authors
The manuscript provides a social cost analysis of alcohol use in the Czech Republic with the major cost areas considered being lost work place productivity, health care and law enforcement. The authors outline the difficulties of assembling administrative data and attributing costs and the underpinning assumptions.
Response: Thank you for your comment on our manuscript.
Introduction
I believe that the paragraph on COVID-19 is unnecessary and can be deleted (also the opening to the abstract)
Response: Thank you for pointing this out. The Introduction and abstract have been changed according to your recommendations, thus covid -19 parts were deleted.
Page 1 line 38: treatment programs are for those with alcohol use problems or disorders: prevention programs are to reduce the uptake of substance use or to prevent transition to problematic substance use.
Response: Thank you for pointing this out, we have accordingly modified the (Page 2, lines 47 – 48)
Page 1 line 38/39 Just clarify that alcohol-caused harms occur across the spectrum of use – including harms to non-users.
Response: Thank you, we agree and have modified the second paragraph in Introduction, (Page 1, lines 39 – 45.
Page 2 lines 46-86 – please split into paragraphs
Response: Thank you for your recommendation, we agree and have, accordingly, split the lines 51 – 99 (previously 46 - 86) into four separate paragraphs.
Page 2 line 52 (and elsewhere) you need to either descript the results of the study and then add the reference or put in the name of at least one author e.g. “Cost related to mental disorders was estimated by Angelis et al [3] ”
Response: Thank you, all citations were modified according to your recommendations.
Page 2: My understanding is that authors can limit social cost studies to any domain, so I’m not sure that this is a criticism of the social cost method. Perhaps just use this section to illustrate the diverse range of costs that have been explored?
Response: Thank you for your comment. Page 2 illustrates the diverse range of costs that were taken into account in previous studies. There are a lot of cost of illness studies that consider different types of cost. That inconsistency within cost of illness studies make it impossible to compare the results of such studies.
Page 3 105. I thought the frictional period was the time until a worker could be replaced (or is that only in cases of death). What about a worker who is temporarily replaced while they are sick.
Response: Thank you for your suggestion, we have checked the paragraph (Page 3, lines 118-122) and modified the text to make the meaning of sentences more clear:
“The value of indirect costs is approximated by the average income of individuals in the future. Thus, this friction period represents the period during which productivity of one worker is lost due to a disease (or death) and the company is looking for the worker’s replacement.”
Method
Page 6 line 264 “Czech Social Security Administration. Sickness register”. Who does this register cover? Only those in full-time employment? Self-employed, “gig” (informal economy)? Presumably no value to voluntary work or housework? Is there anyway of valuing these other forms of lost productivity?
Response: Thank you for your suggestion. Registry of sick leaves in the Czech Republic provide data on employees and self-employed people working in the Czech Republic. In this registry, there are also people who pay their social insurance (even if they´re not working). The database brings the information on individual sick leaves in 2017. Each sick leave case is defined by the age of person, gender, region where he/she works/lives/visits doctor, work type, start and end date of sick leave, the diagnosis related to sick leave. It would have been interesting to explore the voluntary work or housework. However, we didn´t consider this kind of cost, regarding the main subject of the study – generalisation of approaches/methods. We modified the manuscript according to your suggestion and explain who is covered by this register (page 7, lines 307 - 316)
Table 1 – you include Q86.0 FAS – yes caused by alcohol, but most studies report that they are unable to estimate the AF due to poor / missing data on the prevalence of alcohol use by women during pregnancy. Did you manage to find these data?
Response: Thank you for your suggestion. We did extensive research on attributable fractions and we manage to find it in cost of alcohol study published in 2019 in the Czech Republic by Mlčoch et al.(2019) but also in the foreign studies such as Shield et al. (2013), Rehm et al. (2010), Webster et al. (2019), Jones and Belis (2013) Page 8, Table 1
Mlčoch, T.; Chadimová, K.; Doležal, T.; Dolejší, D.; Hájíčková, B.; Mazalová, M.; Lamblová, K. Společenské Náklady Konzumace Alkoholu v České Republice IHETA Autorský Kolektiv; 2019 Shield, K., Parry, C., & Rehm, J. (2013). Chronic Diseases and Conditions Related to Alcohol Use. Alcohol Research : Current Reviews, 35, 155–173.
Webster, L., Angus, C., Brennan, A., Gillespie, D., Court, R., & Street, R. (2019). Smoking attributable fractions for adult diseases in England February 2019. February, 1–31. https://www.sheffield.ac.uk/polopoly_fs/1.828724!/file/ScHARR_AAFs.pdf
Rehm, J., Baliunas, D., Borges, G. L. G., Graham, K., Irving, H., Kehoe, T., Parry, C. D., Patra, J., Popova, S., Poznyak, V., Roerecke, M., Room, R., Samokhvalov, A. V., & Taylor, B. (2010). The relation between different dimensions of alcohol consumption and burden of disease: An overview. Addiction, 105(5), 817–843. https://doi.org/10.1111/j.1360-0443.2010.02899.x
Jones, L., & Bellis, M. A. (2013). Updating England-Specific Alcohol-Attributable Fractions. http://allcatsrgrey.org.uk/wp/download/public_health/alcohol/24892-ALCOHOL-FRACTIONS-REPORT-A4-singles-24.3.14.pdf
Previous social cost of alcohol include estimates of the protective effects of alcohol for some condition. While the extent of these protective effects is increasingly contested (Chikritzhs, T., Stockwell, T., Naimi, T., Andreasson, S., Dangardt, F., Liang, W., 2015. Has the leaning tower of presumed health benefits from ‘moderate’ alcohol use finally collapsed? Addiction 110, 726-727: Sherk, A., Gilmore, W., Churchill, S., Lensvelt, E., Stockwell, T., Chikritzhs, T., 2019. Implications of cardioprotective assumptions for national drinking guidelines and alcohol harm monitoring systems. International Journal of Environmental Research and Public Health 16, 4956) there are some conditions (e.g. diabetes type 2 women only: Ischaemic heart disease) that are generally still included. I couldn’t work out how you addressed “protective effects”.
Response: Thank you for your suggestion. It would have been interesting to explore this aspect however, in our manuscript we consider only cost related to alcohol addiction. We didn´t take into accout positive effects, because there is inconsictency found regarding the positive effects. First of all, not all authors admit including positive effects calculations. Please let me kindly citate Chikritzhs et al. (2015):
„We recommend that future estimates of the alcohol‐related burden of disease and national drinking guidelines should no longer assume any protective effects from low dose consumption. We include in this the Global Burden of Disease estimates, as these can play a major role in either perpetuating the status quo or reforming the field. Guidelines should discourage drinking for health‐related reasons. Health professionals should not recommend moderate alcohol consumption as a means of reducing cardiovascular risk for patients. At the policy level, the hypothesis of health benefits from moderate drinking should no longer play a role in decision making.”
Moreover, nowadays there is an uncertainty in the daily drinking limit that would result in positive effects of drinking. As stated by Sherk et al. (2019) there is a need of downward revision (to 10 – 15 g of pure alcohol per day) of consumption levels resulting in ‘no added’ risk from drinking. For ex. in Canada the limit is 20 g of pure alcohol per day (g/day), in Canada limit for men is 28.8 g/day and limit for women is 19.2 g/day. The authors state, there are differences in drinking limit per day even within Europe for ex. in France (14.4 g/day) and UK (16 g/day).
However, we consider positive effects in the discussion section (Page 28, line 979)
Law enforcement: I appreciated the difficulty of developing an “AF” from these data. However, please give your reader more information on how Zabransky et al. arrived at their figure.
Response: Thank you for your recommendations, we modified our manuscript regarding your recommendations and added the most important information on the estimation of cost related to law enforcement (types of models and framework), that allow to calculate the AF in the study of Zabransky et al. (Page 13, lines 464 -471)
What about the costs of prisons?
Response: Thank you for your suggestion. We considered adding the calculation of costs of prisons, however, in the Czech prison´s statistics the data are aggregated (including alcohol + tobacco, illegal drugs and gambling) and it is not possible to obtain data separately for alcohol. That is the reason why we did not include the cost of prisons (supposing it might be the case in other countries also). However we made modification to discussion section and considered cost of prisons there (Page 24 , lines 839 - 841)
(Minor point – generally contractions (e.g. didn’t) should not be used)
Response: Thank you for pointing this out. We checked the manuscript and replaced all the contractions by full form.
Table 4 and elsewhere. Consider including an online table with your main results converted to USD or Euro for international readers – there are online purchasing power parity calculators to help with that, rather than just using the exchange rate.
Response: Thank you for your recommendation. We agree and converted every sum in CZK to USD (PPP).
Table 11 and section 2.2.5. I thought that taxes and duties were transfer costs and weren’t included in social cost studies. I think that the audits are eligible for inclusion, but perhaps include a reference in 2.2.5 to earlier studies that have included these lost payments?
Response: Thank you for pointing this out, we agree and added the reference in 2.2.5 according to your recommendations. However, in Table 11, there is the tax leakage (tax evasion) identified during the audits of Financial Administration, this is not the official revenue of the country (only potential). This is the money, that should be paid to state (as tax or duty) but it is not. This is the “money” that was only “accidentally revealed” during the audits or by police (it is supposed to be the result of illegal activity/crime, it should be paid to state but it is not, to this fact it is a loss). Therefore, we consider this type of cost to be eligible for our estimation. (lines 556, 744 - 746)
Minor point line 621 typo “More than 180 accidents with 203 SEVERE injuries”
Response: Thank you for pointing this out, we made the correction to “severe”. (line 754)
Minor point line 647 Typo “Total cost related to alcoholtotal cost related to alcohol”
Response: Thank you for pointing this out, we made the correction, there was an issue with the title of Table 15 (accidentally given whole title of Table 15) and this was the reason why the text was duplicated. (line789)
Discussion
“The lost productivity attributable to morbidity is lower than the lost productivity attributable to mortality, what arises from the fact that morbidity is only temporary”. This ignores morbidity that results in long term disability or impairment (which is very hard to value)
Response: Thank you for your suggestion. We agree with you, there might be morbidity that results in long term disability or impairment. If there is any kind of disability, it is supposed, people will be given disability pension (covered in social care cost). However, we only considered the morbidity in 2017 since our morbidity database was limited and covers only temporary sick leaves in 2017. (It includes the information on sick leaves of working people who are employed or self-employed and people who pay social insurance voluntarily). Unfortunately, we do not know whether the morbidity has started longer time before 2017 or it will last until february 2018.
The sick leave is limited to specific period of time. If the sickness continues even above this limit, people are reconsidered to “disable” or to have some kind of disability (regarding to their diagnoses). Generally, people with some kind of disability are given a disability pensions, considered in Social care cost estimation.
“The calculation of attributable fractions from the data of morbidity might take place in future studies. The attributable fractions calculated from morbidity data would be applied to estimate the cost related to social care” Separate alcohol morbidity AF are available. See appendices to Collins DJ, Lapsley HM. The costs of tobacco, alcohol and illicit drug abuse to Australian society in 2004/05. Canberra: Commonwealth of Australia; 2008.
Response: Thank you for pointing this out. We agree there are studies who have already considered the morbidity AF. However, in presented study we did the calculations taking into account fractions attributable to alcohol related mortality as presented also in study of Zabransky et al.(2011)], Mlčoch et al. (2019). In our future studies we would like to do the calculations taking into account fractions attributable to alcohol related to morbidity.We have modified the manuscript to make the meaning of sentence clear to readers.
Mlčoch, T., Chadimová, K., Doležal, T., Dolejší, D., Hájíčková, B., Mazalová, M., & Lamblová, K. (2019). Společenské náklady konzumace alkoholu v České republice iHETA Autorský kolektiv. http://www.iheta.org/ext/publication/files/Report_merged_grant_alkohol_2019-04-10 - final.pdf
Zabransky, T., & Belackova, V. (2011). Společenské náklady užívání alkoholu, tabáku a nelegálních drog v ČR v roce 2007 [Social costs of the use of alcohol, tobacco, and illegal drugs in the Czech Republic, 2007] Central Asia Drug Action Programme (CADAP), Phase 5 View project General Populati. www.adiktologie.cz
Line 761 “or a total ban on the consumption and sale of alcoholic beverages [61-63].” Given the failures of alcohol prohibition in the USA and the devastating effects of the “war on drugs” There may be unintended consequences and costs from this suggestion.
Response: Thank you for your suggestion. We agree with you. However, in 2012, there was partly prohibition of alcohol in the Czech Republic. Prohibition was the subject of the study of Belackovat et al. (2017). As stated by this study: “partial prohibition of sales of > 20% liquors during a methanol outbreak crisis can reduce their consumption, especially among those with more problematic drinking patterns.“. To this fact we modified manuscript and replaced the word TOTAL by PARTIAL (line 1016).
Belackova, V., Janikova, B., Vacek, J., Fidesova, H., & Miovsky, M. (2017). “It can’t happen to me”: Alcohol drinkers on the 2012 outbreak of methanol poisonings and the subsequent prohibition in the Czech Republic. Nordic Studies on Alcohol and Drugs, 34(5), 385–399. https://doi.org/10.1177/1455072517733597
Social cost studies are entitled to draw the parameters of their calculation where ever they please. However, I think that even if you don’t estimate the cost, you should at least acknowledge the extent of harms to others for alcohol use. Other than traffic crashes, I don’t think that these will appear in your costing. While not economic costing studies, alcohol is the only substance thought to cause more harm to others than to the consumer (Bonomo Y, Norman A, Biondo S, Bruno R, Daglish M, Dawe S, et al. The Australian drug harms ranking study. Journal of Psychopharmacology. 2019;33(7):759-68. Nutt DJ, King LA, Phillips LD. Drug harms in the UK: A multicriteria decision analysis. Lancet. 2010;376(9752):1558-65
Response: Thank you for your suggestion, we absolutely agree alcohol causes harm to others. When we first got into the problematic of social cost of alcohol, we were considering other fields and other data types that might be included in our study. In our study we aim to emphasize the need of general methods. When talking about harms to others, the study would be more detailed and more complex what would result in more specific study (not the generalised one). However, after consideration we have extended our discussion regarding your recommendation. (line 842 - 850)
The journal asks reviewers to comment on the adequacy of the literature reviewed – while you include many that I’m unfamiliar with, you might include additional studies from the USA (e.g. Miller TR, Nygaard P, Gaidus A, Grube JW, Ponicki WR, Lawrence BA, et al. Heterogeneous costs of alcohol and drug problems across cities and counties in California. Alcohol Clin Exp Res. 2017;41(4):758-68. Sacks JJ, Gonzales KR, Bouchery EE, Tomedi LE, Brewer RD. 2010 national and state costs of excessive alcohol consumption. Am J Prev Med. 2015;49(5):e73-e9. and Canada (e.g Sherk A, Stockwell T, Rehm J, Dorocicz J, Shield KD, Churchill S. The International Model of Alcohol Harms and Policies: A new method for estimating alcohol health harms with application to alcohol-attributable mortality in Canada. J Stud Alcohol Drugs. 2020;81(3):339-51. Sorge JT, Young M, Maloney-Hall B, Sherk A, Kent P, Zhao J, et al. Estimation of the impacts of substance use on workplace productivity: A hybrid human capital and prevalence-based approach applied to Canada. Can J Public Health. 2020:1-10.) The Sorge et al paper focuses on workplace productivity so may be of particular interest to you.
Response: Thank you for your recommendations, we are grateful for your willingness to improve our manuscript. We went through recommended citations carefully and add them to discussion section. We believe all the studies listed will be enriching and will help our study to succeed. (discussion section, lines 815, 997 - 982)
Regarding to InterMAHP citation - In previous research we were considering InterMAHP software as we believe any kind of automatization (for ex. in data collecting, cost estimation etc.) of cost-of-illness methods would improve benchmarking of countries/regions and would help policy makers to create more efficient strategies in fight against addiction.
Reviewer 2 Report
The written English in this manuscript should be considerably improved; there are many grammatical errors and sections of text that are difficult to read. Also, your reporting of large figures is inconsistent with standard reporting in which numbers lager than 999 million are referred to in billions, rather than thousands of millions. i.e. 1000 million would be better referred to as 1 billion.
I am unsure why the introduction begins with discussing covid-19; it does not seem particularly relevant to examining the costs of alcohol consumption to society. Most healthcare systems have faced financial constraints long before the pandemic. Estimating the economic burden of alcohol extends far beyond the covid-19 pandemic, therefore, unless there is a particular hypothesis that the pandemic has affected the costs of alcohol use, I would suggest making the introduction less focussed on covid-19.
Some distinction needs to be made between estimating the costs of excessive alcohol consumption compared to moderate alcohol consumption (within recommended limits). The authors discuss costs of alcohol generally, without specifying whether it is excessive alcohol consumption that they are focussed on. More background to the negative impacts of excessive alcohol consumption would be beneficial in the introduction. The second paragraph of the introduction in which this is discussed is very limited in detail.
Line 52 and line 73/74: study authors are missing for reference 3 and reference 6.
Lines 47 - 54: It is unclear why the studies listed have been highlighted; they do not seem to be related to alcohol. I suggest the authors focus on discussing studies that can be directly compared to their own study.
Paragraph 3 in the introduction is overly long and should be broken up to improve readability.
Line 101: Author names missing
Lines 115-117: willingness to pay is introduced in the middle of a discussion of the human capital approach. It seems ill-fitting to discuss this here; WTP should be discussed on its own and fully explained as the other approaches are. Furthermore, the approach is actually "contingent valuation" not WTP, WTP is what is derived using the contingent valuation method.
I see you go on to explain WTP further into the paper, however, the introduction to WTP should be provided before it is compared to other approaches in the previous two paragraphs.
Line 142: "A WTP has been developed to include utility in the cost calculation." Please revise this sentence, its meaning is unclear.
Be careful in your discussion of the use of WTP in a cost capacity. In the field of health economics, WTP is used to estimate value, or benefit, associated with an action/good/service and is separate to cost. Values of WTP are used to compare against costs to examine the cost-benefit of some action, rather than used as part of a cost calculation. Estimating WTP comes from classical economic theory to estimate the utility of changes of different states of welfare to the individual. Make are you are clear about the concept of "opportunity cost" when discussing WTP as this is really what WTP is estimating (i.e. how much utility from other uses of income an individual is willing to forgo in order to benefit from what is being valued in the WTP exercise), it is not asking how much individuals believe it should cost to achieve a given outcome.
lines 176/177: What other types of healthcare cost are missing from these studies? What is included in "administrative data"?
lines 183 and 184 - missing citations
line 224: "disponable" is not a commonly used word in English, "available" would be a more widely used and understood term. Database is one word (not two separate words).
Section 2.1: It is not clear whether the data on sickness absence provides information on the reason for absence. Line 267/268 states "The healthcare data contains information on how much money is spent on specific diagnoses." but it is not clear whether this is the same data as sickness absence or some other data. Could it please be clarified which "healthcare data" are being referred to here and if sickness absence is not listed by ICD-10 code, how the data on absence is linked to alcohol-attributable diagnosis costs as described in equation 3?
Lines 295/296: It is not clear what is meant by the phrase "and since it connects several areas, it is not a fully economical method". Could this please be re-phrased to make more clear what is meant by an "economical method" and why the connection of several areas prohibits this?
line 305/306and 309: name of cited authors missing
Line 312: You mention using fractions presented in Tables 1 and 2 but no fractions are presented in Table 1. If these conditions are wholly attributable you may want to add a column to state the fraction is 1 for clarity. In both tables it would be beneficial to cite the source of data for each ICD-10 code rather than just as a list at the bottom of the table, so that readers can verify the data for specific ICD-10 codes if desired.
lines 265 and 353: absences from what precisely? From employment?
line 379: can you please define what is meant by the "bottom-up" approach?
lines 383 and 383 - which years do these costs relate to?
line 384 - please cite the source of the 2017 police cost data
Equation 5 - are you safe to assume the alcohol attributable fraction of police costs is consistent between 2007 and 2017? Do you have any data to base this assumption on? Also, it is not clear where the 2007 data comes from.
lines 408-413: when you talk of damage, who/what is this damage to? damage to individuals? to property? Please clarify. Also, you describe how the amount of damage is calculated, but you do not describe how a cost is attributed to this damage? Please clarify where the costs associated with the criminal damage caused are obtained from. Finally, the sentence "In the area of criminal law, we encounter more cost items that have arisen in connection with alcohol" is not very clear, what do you mean by "more cost items"? consider re-phrasing for clarity.
line 433: rather than referring to "previous subchapter", It would be more beneficial to the reader for you to signpost to the exact section by it's number
line 441: "gross income/per habit" - this is not good English phrasing. Do you mean to say gross income per person? "Habit" is not the correct word. Could you cite the source of your data on gross income by age that is used in equation 9?
Section 2.2.5: It is not clear that the audits in which this cost component is based on are all specifically related to alcohol consumption. Greater explanation of how these audits are related to alcohol consumption would be beneficial to help the reader understand this cost. These audits are described as "audits of tax obligation" but it is unclear which tax obligations these refer to. Furthermore, if you are including costs associated with audits of taxes, this is overestimating the cost of alcohol to the state as you have not discounted the revenue (i.e. negative cost) obtained by the state from duties on alcohol. It would be expected that the duties from alcohol sales would contribute (if not entirely cover) these costs of administration, therefore you may be over-inflating costs to the state by not accounting for this revenue stream.
Equation 10: When you refer to "damage" in this equation, is this a monetary value? Damage in English doesn't mean a monetary amount so if you are referring to a monetary value here I would suggest using a different term such as "cost of damage incurred", or explicitly explaining that damage is a monetary measure. It is also not clear that you have data on the costs of each of these accidents, please clarify where the cost measure attributed to each "damage" is obtained from.
Line 511: what is meant by "personal cost" in this context?
Results - first paragraph: you have not stated in your methods how data on hospital and ambulance costs are calculated as attributable to alcohol. This should be clearly outlined in your methods section.
line 554: you state there is evidence of a higher healthcare cost for men than women - have you conducted any statistical tests on these costs to confirm this difference is statistically significant and not just a chance result? In order to say there is evidence of higher costs you should really be able to demonstrate this statistically. Please provide p-values to demonstrate statistical significance.
Tables 3 - 10: You do not need to state the source if it is your calculations. If no source is cited it is assumed to be your own working.
tables 5, 7, 8: it would be useful if you presented the actual monetary cost associated with each ICD-10 chapters well as the %
line 570: this is the first time you refer to a cost in billions. You should report all costs in a similar fashion that are greater than 999 million.
Table 6: It would be useful to remind readers what each type of disability pension refers to in the table legend.
line 610: some presentational errors with the figure presented in this line, please check this
Fire service cost: it is unclear what "personal cost of fire and rescue services" refers to. Who is this personal to? The fire service? individual tax payers? Please clarify exactly what is meant and who bears this cost
Table 13 and its discussion would be better placed with other social care costs (i.e. after Table 8)
line 647: first 5 words of sentence are repeated
Line 668: It is not clear that this sentence is referring to the calculation of morbidity costs. Do you mean to say that the alcohol attributable fractions for morbidity are taken from those of mortality found in previous studies? If so, could you re-phrase to make this clear.
line 692: which methods did you consider specifically? And which did you decide to use in your study?
Paragraph starting at line 679 - This paragraph is hard for the reader to navigate. It is unclear what the point of the comparison with the studies by Mlčoch et al and Varney & Guest is. I would recommend re-drafting this paragraph and ensure there is a clear point to the study comparisons. What is the upshot of the differences in the approaches that those authors took compared to your approach? How do the approaches taken by those other authors match up with the "variety of applied approaches" referred to in line 694 and how does your "general" approach (line 693) compare?
line 719 - check formatting of figure - "," used instead of "."
Paragraph starting on line 724: it is unclear what the purpose of this paragraph is. How do the methods employed by Mielecka-Kubien et al relate to what was done in your study? Unless some relevant point of comparison is made this paragraph could be removed. It also reports unnecessary level of detail and could be written more concisely.
Your discussion would benefit from a stronger focus. It is difficult to follow your argument throughout. You should focus on how the methods you have employed in your study improve on those employed in the previous studies mentioned. It is not clear from your discussion how your study adds to the literature from a methodological viewpoint, although this seems to be the argument you are trying to convey. Try to integrate comparative studies in a more concise and clearly focussed manner; use particular studies to highlight and support your argument of methodological improvement, rather than just describing in turn the different approaches used in other studies.
Keep your discussion of your study's limitations in one section rather than dispersed throughout your discussion. You could use a "strengths and limitations" subheading within your discussion to highlight this section.
As mentioned in a comment related to your methods, I think it would be prudent to include a discussion of the revenues from alcohol duties. Whilst you may not be able to determine the proportion of these revenues that are used to offset other costs, it is highly likely that at least part of the revenue would be used to contribute towards some of the public sector costs (healthcare, fire department, policing etc) included in your cost calculation. Therefore, your estimated cost is likely to be over-estimating the true social cost without accounting for this. If you view alcohol duty revenue as a transfer cost, and therefore exclude it, this should also be explained in your discussion.
Your discussion would also benefit from some consideration of whether you believe your estimate to be an over- or under-estimate of the true cost of alcohol use, based on the data you know to be missing and the assumptions made in your calculations.
Your conclusion should include a summary of your key finding i.e. the total social cost of alcohol use that your study has estimated.
Your conclusion states "The study’s results represent a valuable platform for the creators of strategic health plans and also policies’ makers" but it is not clear to the reader why this is the case. How could your results be used by these institutions? Is the important result your cost estimate or the attempt at providing a generalisable approach to measuring costs?
Author Response
Dear Reviewer,
Thank you very much for the careful and thorough reading of our manuscript and for the thoughtful comments and constructive suggestions. We hope that these revisions improve the paper such that you now deem it worthy of publication in „International Journal of Environmental Research and Public Health“ and our revision has improved the paper to a level of your satisfaction. We have carefully revised the paper and rewrite accordingly. In the following we respond in detail to the all comments. All changes in the revised manuscript are highlighted using the "Track Changes" function.
Thank you for giving us the opportunity to revise our manuscript. We would like to send our revised study again for considering and we look forward to hearing from you.
Kind regards,
Beata Gavurova and Miriama Tarhanicova
Open Review
Comments and Suggestions for Authors
The written English in this manuscript should be considerably improved; there are many grammatical errors and sections of text that are difficult to read. Also, your reporting of large figures is inconsistent with standard reporting in which numbers lager than 999 million are referred to in billions, rather than thousands of millions. i.e. 1000 million would be better referred to as 1 billion.
Response: Thank you for your recommendation. The manuscript has been checked and revised in terms of grammar and writing style. English language was improved. We also checked large figures (larger than 999 million) and modified it to billion.
I am unsure why the introduction begins with discussing covid-19; it does not seem particularly relevant to examining the costs of alcohol consumption to society. Most healthcare systems have faced financial constraints long before the pandemic. Estimating the economic burden of alcohol extends far beyond the covid-19 pandemic, therefore, unless there is a particular hypothesis that the pandemic has affected the costs of alcohol use, I would suggest making the introduction less focussed on covid-19.
Response:Thank you for pointing this out. We agree and have modified the introduction according to your recommendation.
Some distinction needs to be made between estimating the costs of excessive alcohol consumption compared to moderate alcohol consumption (within recommended limits). The authors discuss costs of alcohol generally, without specifying whether it is excessive alcohol consumption that they are focussed on. More background to the negative impacts of excessive alcohol consumption would be beneficial in the introduction. The second paragraph of the introduction in which this is discussed is very limited in detail.
Response: Thank you for your recommendation. We agree and modified the introduction regarding your recommendation. (Page 5, line 220)
We also agree with your recommendation regarding the negative impacts of excessive alcohol consumption and modified manuscript according to your recommendation..(Page 1, lines 39 -45)
Line 52 and line 73/74: study authors are missing for reference 3 and reference 6.
Response:Thank you, the authors for reference 3 (currently line 61) and 6 (currently line 63/64) were added.
Lines 47 - 54: It is unclear why the studies listed have been highlighted; they do not seem to be related to alcohol. I suggest the authors focus on discussing studies that can be directly compared to their own study.
Response: Thank you for your suggestion. In presented study we emphasized the need of general methods and approaches that are easily applicable in cost of illness study. Highlighted studies were listed because we would like to point out heterogeneity of cost of illness studies in general. (lines 56 - 64)
Paragraph 3 in the introduction is overly long and should be broken up to improve readability.
Response: Thank you for your suggestion. To improve readability the paragraph 3 in the introduction was splitted into four paragraphs. (Page 2, lines 51 - 99)
Line 101: Author names missing
Response: Thank you for pointing this out, the author names were added (Page 3, lines 115-116)
Lines 115-117: willingness to pay is introduced in the middle of a discussion of the human capital approach. It seems ill-fitting to discuss this here; WTP should be discussed on its own and fully explained as the other approaches are. Furthermore, the approach is actually "contingent valuation" not WTP, WTP is what is derived using the contingent valuation method.
Response: Thank you for your recommendations. Our manuscript was modified accordingly to your recommendations.(Page 3-4, lines 143 - 167)
I see you go on to explain WTP further into the paper, however, the introduction to WTP should be provided before it is compared to other approaches in the previous two paragraphs.
Response: Thank you for your recommendation, we agree and replaced the paragraph above. (Page 3-4, lines143 - 167 )
Line 142: "A WTP has been developed to include utility in the cost calculation." Please revise this sentence, its meaning is unclear.
Response: Thank you for pointing this out. We have revised the sentence and rewrite it to following form:
“It introduces the concept of utility in the cost calculation.” (Page 3, lines 164-165)
Be careful in your discussion of the use of WTP in a cost capacity. In the field of health economics, WTP is used to estimate value, or benefit, associated with an action/good/service and is separate to cost. Values of WTP are used to compare against costs to examine the cost-benefit of some action, rather than used as part of a cost calculation. Estimating WTP comes from classical economic theory to estimate the utility of changes of different states of welfare to the individual. Make are you are clear about the concept of "opportunity cost" when discussing WTP as this is really what WTP is estimating (i.e. how much utility from other uses of income an individual is willing to forgo in order to benefit from what is being valued in the WTP exercise), it is not asking how much individuals believe it should cost to achieve a given outcome.
Response:Thank you for your detailed explanation on willingness to pay approach. We are really gratefull for this comment as it enrichement is evident. The paragraph that takes into consideration willingness to pay approach was modified regarding your recommendations. (Page 3, lines 164-165)
lines 176/177: What other types of healthcare cost are missing from these studies? What is included in "administrative data"?
Response: Thank you for your suggestion. The administrative data usually includes the general expenditure of healthcare providers covering overhead charges. There are the cost related to treatment of specific diagnosis or the diagnostic procedures missing (such as cost of magnetic resonance, CT, X-ray etc.). The manuscript was enriched by this information (Page 5, lines 216 - 219)
lines 183 and 184 - missing citations
Response: Thank you, the authors for references in lines 183 and 184 were added (Page 5, lines 225 - 226)
line 224: "disponable" is not a commonly used word in English, "available" would be a more widely used and understood term. Database is one word (not two separate words).
Response:Thank you for pointing this out. We have checked the text and correct both words mentioned. (Page 6, line 266)
Section 2.1: It is not clear whether the data on sickness absence provides information on the reason for absence. Line 267/268 states "The healthcare data contains information on how much money is spent on specific diagnoses." but it is not clear whether this is the same data as sickness absence or some other data. Could it please be clarified which "healthcare data" are being referred to here and if sickness absence is not listed by ICD-10 code, how the data on absence is linked to alcohol-attributable diagnosis costs as described in equation 3?
Response: Thank you for pointing this out. When talking about healthcare data – that includes the cost related to individual diagnoses. When someone visit ambulance because of illness (for ex. related to diagnosis J00) there are some cost related to this visit (work of a doctor, a nurse). When someone is hospitalised (for ex. related to diagnosis C00), there are some cost related to this hospitalisation (work of a doctor, a nurse, medication, CT, magnetic resonance etc.). We firstly identified all diagnoses attributable to alcohol (Table 1 and Table 2) and then got the data on cost for each diagnosis related to alcohol
Data on sickness absence covers the information on sick leaves regarding the diagnosis (related to sick leaves). If someone is working and suddenly get sick (he´s sickness is caused by diagnosis J00), he has to recover at home for 2 weeks – so two weeks presents “lost productivity” to his employer and to society as well (recovering he is not able to produce GDP). We did not consider replacing those people. Every case in Registry of sickness absence contains the information on age, sex, region, start date of sickness, end date of sickness and diagnosis related to this sickness. Therefore, it was possible to add the attributable fraction to every case of sickness absence. (Page7, lines 306 - 326)
Lines 295/296: It is not clear what is meant by the phrase "and since it connects several areas, it is not a fully economical method". Could this please be re-phrased to make more clear what is meant by an "economical method" and why the connection of several areas prohibits this?
Response: Thank you for pointing this out, we agree and re-phrased the paragraph according to your recommendation as follows:
“It is used to estimates the cost of illness (or specified issue). Illness or specified issue (such as alcohol addiction) have severe consequences in many areas of life (law enforcement, healthcare, economics, society, etc.). When applying this method, firstly, it is necessary to gain insight into those consequences and their relation to individual areas of life, therefore the knowledge of estimation process (methods and approaches) is not sufficient.” (Pages 7-8, lines 348 -354)
line 305/306and 309: name of cited authors missing
Response: Thank you for pointing this out, the author names were added (Page 8 ,lines 366, 368)
Line 312: You mention using fractions presented in Tables 1 and 2 but no fractions are presented in Table 1. If these conditions are wholly attributable you may want to add a column to state the fraction is 1 for clarity. In both tables it would be beneficial to cite the source of data for each ICD-10 code rather than just as a list at the bottom of the table, so that readers can verify the data for specific ICD-10 codes if desired.
Response: Thank you for pointing this out. All recommendations have been accepted and included in the revised manuscript. We added the attributable fractions to Table 1 and added the references to each ICD – 10 code.(Page 8 – 9, lines 376, 380)
lines 265 and 353: absences from what precisely? From employment?
Response: Thank you for pointing this out. We checked this issue in manuscript and decide to describe Registry of sick leaves in the Czech Republic precisely. This registry provides data on sick leaves of employees, self-employed people working in the Czech Republic and people who pay the social insurance voluntarily. The manuscript was modified to provide missing information clearly to its readers. (lines 307 - 315, 429 - 435)
line 379: can you please define what is meant by the "bottom-up" approach?
Response: Thank you for your recommendation. Normally the bottom-up approach is applied when we know the average cost per unit and then there is a generalisation of this cost to study population. However, in this part of study the top-down approach was applied as there are fractions attributable to alcohol calculated and used to estimate the cost related to alcohol. (Page 13, line 461)
lines 383 and 383 - which years do these costs relate to?
Response: Thank you for pointing this out. We add the years these costs relate to. (Page 13, lines 464, 472)
line 384 - please cite the source of the 2017 police cost data
Response:Thank you for your suggestion, however there is the citation for the data of Police, we modified the citation from Police [54] to Police of the Czech Republic [54] to make it more clear. This data were taken from the official statistics of the Police of the Czech republic:
“Police of the Czech Republic, Statistické Přehledy Kriminality Za Rok 2017 - Policie České Republiky; 2018” (Page 14, line 506)
Equation 5 - are you safe to assume the alcohol attributable fraction of police costs is consistent between 2007 and 2017? Do you have any data to base this assumption on? Also, it is not clear where the 2007 data comes from.
Response: Thank you for your suggestion, we are grateful for this comment as it points to an important aspect of data consistency. However, we are absolutely safe about assumption of consistency between 2007 and 2017. It was communicated with experts who deal with alcoholic issues regarding the police. The examined data were published in Ministry of Internal Affairs [40] (Page 13, ine 471)
lines 408-413: when you talk of damage, who/what is this damage to? damage to individuals? to property? Please clarify. Also, you describe how the amount of damage is calculated, but you do not describe how a cost is attributed to this damage? Please clarify where the costs associated with the criminal damage caused are obtained from. Finally, the sentence "In the area of criminal law, we encounter more cost items that have arisen in connection with alcohol" is not very clear, what do you mean by "more cost items"? consider re-phrasing for clarity.
Response: Thank you for your recommendation, we agree and modified the manuscript to make “damage issue” clear. Usually this damage presents the property damage to its owner. This damage is estimated by experts who usually come to place where the crime was committed and is listed in the official public crime statistics of Police.(Page 14, lines 497 - 506)
Thank you also for your suggestion regarding the sentence: “In the area of criminal law..”, We agree and replaced “more cost items” by “cost related to different property damages” (Page 14, line 501)
line 433: rather than referring to "previous subchapter", It would be more beneficial to the reader for you to signpost to the exact section by it's number
Response: Thank your for pointing this out. We checked it, and replace the “previous subchapter” by chapter 2.2 Methods.(Page 14, line 534)
line 441: "gross income/per habit" - this is not good English phrasing. Do you mean to say gross income per person? "Habit" is not the correct word. Could you cite the source of your data on gross income by age that is used in equation 9?
Response: Thank you for pointing this out. We modified manuscript according to your recommendation and replaced the word “habit” by “person. We also add the reference to gross income, that comes from official Czech Statistics Office (Page 15, line 548)
Section 2.2.5: It is not clear that the audits in which this cost component is based on are all specifically related to alcohol consumption. Greater explanation of how these audits are related to alcohol consumption would be beneficial to help the reader understand this cost. These audits are described as "audits of tax obligation" but it is unclear which tax obligations these refer to. Furthermore, if you are including costs associated with audits of taxes, this is overestimating the cost of alcohol to the state as you have not discounted the revenue (i.e. negative cost) obtained by the state from duties on alcohol. It would be expected that the duties from alcohol sales would contribute (if not entirely cover) these costs of administration, therefore you may be over-inflating costs to the state by not accounting for this revenue stream.
Response: Thank you for your suggestion, we modified the manuscript (Section 2.2.5) according to your suggestion. We re-phrased the lines 554 – 588, 744 - 749 to make it more clear to readers. In our study we only consider cost as it is cost of illness study, therefore we do not consider benefit (for ex. revenue). We understand that taxes/duties represent the revenues to states. Therefore, we only consider the cost that relates to administration of taxes and the tax audits and the amount that arises from the illegal activity/crime (tax leakage – tax evasion). Officially this money is not part of the revenues as it wasn’t administered officially. Tax leakage cost/tax evasion cost is a result of a crime/illegal activity. However, it would be interesting to explore more on relation between cost of illness study and benefit analysis in future studies.
Tax administration was also estimated in study of Mlčoch et al. (2019)
Mlčoch, T.; Chadimová, K.; Doležal, T.; Dolejší, D.; Hájíčková, B.; Mazalová, M.; Lamblová, K. Společenské Náklady Konzumace Alkoholu v České Republice IHETA Autorský Kolektiv; 2019 Shield, K., Parry, C., & Rehm, J. (2013). Chronic Diseases and Conditions Related to Alcohol Use. Alcohol Research : Current Reviews, 35, 155–173.
Equation 10: When you refer to "damage" in this equation, is this a monetary value? Damage in English doesn't mean a monetary amount so if you are referring to a monetary value here I would suggest using a different term such as "cost of damage incurred", or explicitly explaining that damage is a monetary measure. It is also not clear that you have data on the costs of each of these accidents, please clarify where the cost measure attributed to each "damage" is obtained from.
Response: Thank you for pointing this out and helping us to find out correct meaning of word “damage”. We modified manuscript according to your recommendation. Data on “damage” are obtained from official Statistics of police concerning traffic accidents (The Police of the Czech Republic [56]) (line 602-603)
Line 511: what is meant by "personal cost" in this context?
Response: Thank you for pointing this out, we checked it and modified to make it more clear to readers. (lines 631 -632)
Results - first paragraph: you have not stated in your methods how data on hospital and ambulance costs are calculated as attributable to alcohol. This should be clearly outlined in your methods section.
Response: Thank you for your pointing this out. We agree, the presentation of method applied to healthcare cost estimation was not clear. We checked our manuscript and modified chapter 2.2.1 Healthcare cost (lines 408 - 423)
line 554: you state there is evidence of a higher healthcare cost for men than women - have you conducted any statistical tests on these costs to confirm this difference is statistically significant and not just a chance result? In order to say there is evidence of higher costs you should really be able to demonstrate this statistically. Please provide p-values to demonstrate statistical significance.
Response: Thank you for pointing this out. The p-values were added to both – hospital and ambulance cost. (lines 677 - 682)
Tables 3 - 10: You do not need to state the source if it is your calculations. If no source is cited it is assumed to be your own working.
Response:Thank you for your recommendation, the sources were deleted from all tables.
tables 5, 7, 8: it would be useful if you presented the actual monetary cost associated with each ICD-10 chapters well as the %
Response: Thank you for your recommendation, the actual monetary cost associated with each ICD-10 chapters were added to tables 5,7,8.
line 570: this is the first time you refer to a cost in billions. You should report all costs in a similar fashion that are greater than 999 million.
Response: Thank you for your recommendations, we checked it and modified large figure (larger than 999 million) to billion.
Table 6: It would be useful to remind readers what each type of disability pension refers to in the table legend.
Response: Thank you for pointing this out. We agree and add the description in table legend (lines 700-701):
“*decline in working capacity by 35 – 49%, ** decline in working capacity by 50 – 69%, ***decline in working capacity by 70% or more”
line 610: some presentational errors with the figure presented in this line, please check this
Response: Thank you for pointing this out. The first column in the tables presents the total cost that are related to law enforcement. However, we decided to delete the first column to make the table clear to readers. (Table 9,line 730)
Fire service cost: it is unclear what "personal cost of fire and rescue services" refers to. Who is this personal to? The fire service? individual tax payers? Please clarify exactly what is meant and who bears this cost
Response: Thank you for pointing this out, we modified the manuscript and clarified what exactly is meant by fire service cost. (lines 762 - 767)
Table 13 and its discussion would be better placed with other social care costs (i.e. after Table 8)
Response: Thank you, we are grateful for your suggestion but we agree with you only partly, therefore we decided not to change placement of cost related to social care services. We would like to kindly explain our decision. In the social care cost there are the cost related to disability and morbidity. On the contrary, social services costs presents the social cost that is related to alcohol, but is not primarily related to any kind of morbidity or physical disability.
line 647: first 5 words of sentence are repeated
Response: Thank you for pointing this out, we made, accordingly, the correction. (line 789)
Line 668: It is not clear that this sentence is referring to the calculation of morbidity costs. Do you mean to say that the alcohol attributable fractions for morbidity are taken from those of mortality found in previous studies? If so, could you re-phrase to make this clear.
Response: Thank you for pointing this out, we agree and modified the manuscript to make this clear. We modified the discussion part, therefore we hope to clearly find the sentence meaning. (lines 829 -830)
line 692: which methods did you consider specifically? And which did you decide to use in your study?
Response: Thank you for pointing this out. We decided to add information on the methods in discussion part as follows:
Before conducting this study, we aimed to apply friction cost method and willingness to pay approach as both are considered to be also relevant for this kind of study (lines 835 - 841)
The attributable fractions related to mortality was applied in calculation of healthcare cost, social care cost and lost productivity. To estimate the lost productivity related to mortality we applied human capital approach. Top-down approach was applied to estimate healthcare cost, social care cost and law enforcement. Bottom-up approach was applied in calculation of Public Administration cost and traffic accident cost. (lines 829 - 934)
Paragraph starting at line 679 - This paragraph is hard for the reader to navigate. It is unclear what the point of the comparison with the studies by Mlčoch et al and Varney & Guest is. I would recommend re-drafting this paragraph and ensure there is a clear point to the study comparisons. What is the upshot of the differences in the approaches that those authors took compared to your approach? How do the approaches taken by those other authors match up with the "variety of applied approaches" referred to in line 694 and how does your "general" approach (line 693) compare?
Response: Thank you for your suggestion, we modified our manuscript according to your recommendation and redrafted mentioned paragraph. All the comparison made within discussion might be taken into consideration in future studies. As we consider data to be the greatest limitation of cost-of-illness studies, this part provides insights into other methods that might be applied. Discussion part provides the explanation why we consider our approaches/methods as the most appropriate for recent study and future studies as well. Our general approach is easy to apply and does not pose strict requirements for research teams in regards to data. (discussion section, lines 798 -1021)
line 719 - check formatting of figure - "," used instead of "."
Response: Thank you. We replaced “,” by “.”.line 946
Paragraph starting on line 724: it is unclear what the purpose of this paragraph is. How do the methods employed by Mielecka-Kubien et al relate to what was done in your study? Unless some relevant point of comparison is made this paragraph could be removed. It also reports unnecessary level of detail and could be written more concisely.
Response: Thank you for pointing this out, we agree and removed the paragraph regarding your recommendation.
Your discussion would benefit from a stronger focus. It is difficult to follow your argument throughout. You should focus on how the methods you have employed in your study improve on those employed in the previous studies mentioned. It is not clear from your discussion how your study adds to the literature from a methodological viewpoint, although this seems to be the argument you are trying to convey. Try to integrate comparative studies in a more concise and clearly focussed manner; use particular studies to highlight and support your argument of methodological improvement, rather than just describing in turn the different approaches used in other studies.
Response: Thank you for pointing this out. We are grateful for this comment and decided to rephrasing the discussion section in regard to your recommendations (discussion section, lines 798 -1021)
Keep your discussion of your study's limitations in one section rather than dispersed throughout your discussion. You could use a "strengths and limitations" subheading within your discussion to highlight this section.
Response: Thank you for your suggestion. We agree and decide to make the structure of our discussion section more clear to readers, thus we revised the discussion. However, we do not find adding subheading to our discussion important as it is split in many paragraphs. (discussion section lines 798 -1021)
As mentioned in a comment related to your methods, I think it would be prudent to include a discussion of the revenues from alcohol duties. Whilst you may not be able to determine the proportion of these revenues that are used to offset other costs, it is highly likely that at least part of the revenue would be used to contribute towards some of the public sector costs (healthcare, fire department, policing etc.) included in your cost calculation. Therefore, your estimated cost is likely to be over-estimating the true social cost without accounting for this. If you view alcohol duty revenue as a transfer cost, and therefore exclude it, this should also be explained in your discussion.
Response: Thank you for your recommendation, we are grateful for your comment and absolutely agree with you on taxes/duties. Both represent the revenues of the country, therefore it might be used to contribute towards some of the public sector entities (even Financial Administration) affected by alcohol consumption.
However, in our study we consider the cost of tax administration and the cost related to audits of taxpayers. The aim of these audits is to control whether entities (enterprises or individuals) pay tax/duty as they should (in regards to legislation). As we know how many audits were made in relation to alcohol, it was possible to calculate the attributable fraction.
Moreover, we also consider the tax leakage (or the tax evasion) cost – this is the money that should be administered in regards to legislation, but there are entities who make their activity illegal and therefore, they do not pay tax (illegaly). In our study, tax evasion = money that should be the revenues of country but it is not. Instead, it is kind of “loss”. If the estimation of social cost was more detailed and contains the information on “alcohol benefits”, it would not be the estimation of cost anymore but cost-benefit analysis instead. More detailed analysis would not fulfil the assumption of generalised approach.
When considering this type of cost included in our study, we were inspired by Mlčoch et al. (2019) and Zabransky et al. (2011).
Mlčoch, T.; Chadimová, K.; Doležal, T.; Dolejší, D.; Hájíčková, B.; Mazalová, M.; Lamblová, K. Společenské Náklady Konzumace Alkoholu v České Republice IHETA Autorský Kolektiv; 2019 Shield, K., Parry, C., & Rehm, J. (2013). Chronic Diseases and Conditions Related to Alcohol Use. Alcohol Research : Current Reviews, 35, 155–173.
Zabransky, T., & Belackova, V. (2011). Společenské náklady užívání alkoholu, tabáku a nelegálních drog v ČR v roce 2007 [Social costs of the use of alcohol, tobacco, and illegal drugs in the Czech Republic, 2007] Central Asia Drug Action Programme (CADAP), Phase 5 View project General Populati. www.adiktologie.cz
Your discussion would also benefit from some consideration of whether you believe your estimate to be an over- or under-estimate of the true cost of alcohol use, based on the data you know to be missing and the assumptions made in your calculations.
Response: Thank you for your comment. We are grateful for pointing this out as we think this is important issue related to our study. We modified discussion regarding your recommendation.(lines 969 -973)
Your conclusion should include a summary of your key finding i.e. the total social cost of alcohol use that your study has estimated. Your conclusion states "The study’s results represent a valuable platform for the creators of strategic health plans and also policies’ makers" but it is not clear to the reader why this is the case. How could your results be used by these institutions? Is the important result your cost estimate or the attempt at providing a generalisable approach to measuring costs?
Response: Thank you for your recommendation. We modified conclusion to make it more clear to readers and add the total social cost of alcohol use in the Czech Republic in 2017 (discussion section, lines 798 -1021)
We are also grateful for your comment on utilization of our results by institutions. As it is stated, the main aim of our study was to bring the generalisable approach of measuring cost related to alcohol, therefore “the result of our study” are approaches and methods that might be used to estimate cost of alcohol in future studies. However, as we use real data, it also brings the information on how much “the negative consequences of drinking” cost in different fields (healthcare, social care, law enforcement, etc.), therefore policy makers are able to identify precisely where alcohol consequences arise the most. Next, methods and approaches applied in our study might be used by research teams who will cooperate in policy making. If researchers will use the same methods and approaches during longer period (more years), it will be possible to do the comparison of the results and to decide whether applied restrictions (regarding alcohol) have impact on cost of alcohol.
Regarding your suggestion, conclusion has been formulated more clearly. (discussion section, lines 798 -1021)
Round 2
Reviewer 1 Report
All my earlier comments have been addressed and thank you for clarifying the text that I had misinterpreted.
One minor point: in the method you need to say something like " all values are reported in USD, with a purchasing power party (PPP) calculator used to convert Czech krona to USD values for 2017 (??)"
Something has happened to the format of table 2 - it is 38 pages long
Author Response
Dear Reviewer,
Thank you for your recommendation, we are grateful for your willingness to improve the quality of our manuscript. We add the text and citations according to your recommendations. (lines 303 - 305).